# Single-cell analysis reveals critical toxin/antitoxin ratio triggering persister resuscitation

Lina Wu [1,✉], Qingqing Wang [1,3], Xinyi Hong [1,3], Xueer Cai[1], Litinghui Zhang[1], Min Li[1], Mingkai Wu [1], Thomas K Wood[2] & Xiaomei Yan [1,✉]

## Abstract

**Persisters represent a transient, antibiotic-tolerant subpopulation within isogenic bacterial populations, contributing to infection relapses. However, the mechanisms driving persister formation and resuscitation remain elusive. Here, we developed nano-flow cytometry (nFCM)-based methods for single-cell quantification of toxin (T) RelE and antitoxin (A) RelB levels, as well as for monitoring persister states through cell wall growth. We demonstrate that bacteria elevate the T/A ratio through two distinct TA expression modalities to withstand bacteriostatic antibiotic challenge, with $T/A = 1.0$ as a critical threshold. Intriguingly, single-cell resuscitation dynamics revealed that subinhibitory antibiotic exposure promotes entry into a deeper dormant state characterized by elevated T/A ratios, underscoring the importance of maximizing therapeutic antibiotic concentrations. Crucially, we uncovered a triphasic detoxification process during resuscitation where progressive toxin depletion drives T/A ratio reduction to a critical proliferation-permissive threshold. Proteomic profiling unveiled that persisters with high RelE production have increased transmembrane transporter levels linked to stress response and drug efflux. Our findings offer pivotal molecular insights underlying persister transitions and underscore the need for high-throughput, single-cell analysis of these heterogeneity phenotypes.**

**Keywords** Single-cell Analysis; Toxin-antitoxin System; Nano-flow Cytometry; Toxin/antitoxin Ratio; Persister Formation and Resuscitation
**Subject Category** Microbiology, Virology & Host Pathogen Interaction

## Introduction

Persister cells represent a minority phenotype within isogenic bacterial populations capable of transiently withstanding the lethal effects of antibiotics (Dewachter et al, 2019; Gollan et al, 2019; Lewis, 2007). These cells are notorious for their association with chronic and recurrent infections, presenting formidable challenges in clinical treatment (Fang and Allison, 2023; Fauvart et al, 2011; Fisher et al, 2017). Moreover, growing evidence suggests that persisters contribute to the acceleration of antibiotic resistance evolution, thereby exacerbating the ongoing antibiotic crisis (Cohen et al, 2013; Levin-Reisman et al, 2017; Liu et al, 2020; Van den Bergh et al, 2016).

Toxin-antitoxin (TA) systems are widely recognized as pivotal regulators of bacterial persistence (Fernández-García et al, 2024; Harms et al, 2016; Page and Peti, 2016). These systems consist of stable toxins and labile antitoxins, which neutralize toxins to safeguard essential cellular functions (Jurėnas et al, 2022). Perturbations in antitoxin levels, whether through reduced synthesis or increased degradation, can lead to an imbalance favoring the toxic activity of free toxins, resulting in cell death or growth arrest (Harms et al, 2018). Ectopic expression of multiple toxins within TA systems (e.g., HipBA, MazEF, RelBE, TisB/IstR1, and Hok/Sok) induces bacterial entry into the persister state (Dörr et al, 2010; Keren et al, 2004; Pedersen et al, 2002; Schumacher et al, 2009; Verstraeten et al, 2015) whereas inactivation of TA systems like MqsRA, RelBE, or HipBA impairs persister formation to varying degrees (Keren et al, 2004; Kim and Wood, 2010; Wu et al, 2015). Transcriptional analyses have revealed upregulation of multiple TA loci in persisters (Alkasir et al, 2018; Keren et al, 2004; Ronneau and Helaine, 2019). However, due to the fact that only a tiny fraction of cells activate TA systems to become persisters, these phenomena can be overlooked in bulk population studies. To date, the direct involvement of TA systems in persistence at the single-cell level in native contexts remains largely unexplored.

Many TA systems exhibit conditional cooperativity, with type II TA systems—categorized by transcriptional autoregulation and protein-dependent toxin inhibition—predominating to engender phenotypic bistability (Page and Peti, 2016). The core mechanism involves varying the toxin/antitoxin (T/A) ratio to modulate transcription (Garcia-Rodriguez et al, 2024), thereby guiding the transition between growth and persistence (Harms et al, 2018). Mathematical models, informed by biochemical insights and TA complex structures, substantiate the linkage between the persister state and TA systems (Cataudella et al, 2013; Cataudella et al, 2012; Gelens et al, 2013). Theoretical models and ectopic overexpression

[1]Department of Chemical Biology, MOE Key Laboratory of Spectrochemical Analysis & Instrumentation, Fujian Key Laboratory of Chemical Biology (Xiamen University), State Key Laboratory of Physical Chemistry of Solid Surfaces, College of Chemistry and Chemical Engineering, Xiamen University, Xiamen, Fujian 361005, China. [2]Department of Chemical Engineering, Pennsylvania State University, University Park, PA 16802-4400, USA. [3]These authors contributed equally: Qingqing Wang, Xinyi Hong.
✉E-mail: alina1222@xmu.edu.cn; xmyan@xmu.edu.cn

studies highlight the role of toxin HipA in inducing persister formation above a critical threshold, dependent on the levels of its cognate antitoxin HipB (Rotem et al, 2010). Recent hypotheses propose that T/A ratio dynamics dictate bacterial entry into and exit from the persister state (Page and Peti, 2016; Rotem et al, 2010), advocating for concurrent investigation of both TA components to deepen our understanding of bacterial persistence.

This is challenging for several reasons. First, the transient nature and heterogeneity of persister cells (Goormaghtigh and Melderen, 2019) make it difficult to dissect how TA systems precisely mediate persister formation and awakening using conventional bulk assays. Second, the low abundance of toxin and antitoxin poses significant challenges for sensitive and quantitative measurement at the single-cell level in analytical approaches. Third, the rarity of persisters (approximately one in $10^5$ of the bacterial population) necessitates sampling large numbers of cells. Flow cytometry (FCM) offers a high-throughput technique for multi-parameter single-cell measurement. However, the low-level expression of most TA system present significant challenge for quantitative measurement of TA expression in their native state (Goormaghtigh et al, 2018; Schumacher et al, 2015).

Employing strategies for single-molecule fluorescence detection in a sheathed flow, our laboratory developed a nano-flow cytometer (nFCM) that enables light scattering detection of single viruses, with a fluorescence detection limit as low as three Alexa Fluor 532 molecules (Ma et al, 2016; Niu et al, 2021; Zhu et al, 2014; Zhu et al, 2010). The superior sensitivity of nFCM in simultaneous fluorescence and light scattering measurement offers a unique advantage for the quantitative assessment of TA levels and bacterial states at the single-cell level. The small tetracysteine (TC) tag of the tetracysteine-biarsenical (TC-FlAsH) system facilitates site-specific fluorescent labeling, with minimal interference to protein structure and function (Adams et al, 2002; Hoffmann et al, 2010; Minoshima et al, 2024; Wu et al, 2011; Wu et al, 2019). In this study, we combined the TC-FlAsH system with wheat germ agglutinin fluorescence dye (Alexa Fluor®633-WGA, flWGA) and nFCM to establish methods for the sensitive, high-throughput, single-cell detection of the toxin RelE and antitoxin RelB in their native contexts; i.e., single gene copies in the chromosome under control of the native promoter, as well as for monitoring bacterial persister states. These approaches provide novel insights into the mechanisms underlying TA system function during persister formation and resuscitation by providing the first determination of T and A levels in single cells and by demonstrating that T/A ratios serve as a critical threshold in this process.

# Results

## Quantitative analysis of TA expression in single bacteria using the nFCM-T/A-TC-FlAsH strategy

The principle of TA expression quantification is schematically described in Fig. 1A. We selected the RelBE system, a crucial TA pair in the formation of persisters (Pedersen et al, 2003; Tashiro et al, 2012), as our model. To measure the native expression of toxin RelE and antitoxin RelB, we constructed two E. coli BW25113 strains. These strains harbor a 12-amino acid TC tag fused to the C-terminus of RelE and to the N-terminus of RelB ($RelE^{TC}$ and $RelB^{TC}$, respectively) as the sole copies of each protein at their native chromosomal loci under

control of the native promoter (Fig. 1A). Phenotypic analysis and Data-Independent Acquisition (DIA) proteomics confirmed that the TC tag insertion did not significantly affect bacterial growth (Appendix Fig. S1A,B), persistence (Appendix Fig. S1C), or global proteome homeostasis (Appendix Fig. S1D). The TC-tagged toxin or antitoxin were specifically labeled with the cell membrane-permeant biarsenical dye FlAsH-EDT$_2$ (EDT, 1, 2-ethanedithiol) and analyzed using nFCM. Although the abundance of toxin and antitoxin in single cells is typically low under non-stressed conditions, a distinct right shift in fluorescence burst area was observed for both $RelE^{TC}$ and $RelB^{TC}$ on bivariate dot plots and histograms during exponential growth (Fig. 1B) compared to the negative control (wild-type strain without the TC tag). The median fluorescence intensities (MFIs) for $RelE^{TC}$ and $RelB^{TC}$ were increased by 1.1- and 1.55-fold relative to the negative control, respectively. These results demonstrate that the expression of RelE and RelB can be detected in single cells with high sensitivity using the nFCM-T/A-TC-FlAsH strategy. A T/A ratio of RelE/RelB = 0.18 was calculated based on the MFI of $RelE^{TC}$ versus that of $RelB^{TC}$ after background signal subtraction for each protein (Fig. 1C), i.e., (MFI$_{RelE}{}^{TC}$-MFI$_{wt}$) versus (MFI$_{RelB}{}^{TC}$-MFI$_{wt}$), indicate antitoxin RelB is present at a significantly higher level than RelE. This finding aligns with the established understanding that antitoxins are translated at a higher rate than toxins and fully neutralize toxin activity completely under natural growth conditions (Li et al, 2014). Collectively, these data indicate that the chromosome fusion of TC tags to RelE and RelB accurately and specifically reports their expression. Given the low cellular amounts of these proteins, this represents, to our knowledge, the first detection of TA expression in their native contexts at the single-cell level.

## Bacteria elevate the T/A ratio through distinct expression modes to resist antibiotic stress

Next, we investigated the expression of RelE and RelB in bacteria exposed to varying concentrations of rifampicin (RIF) using the nFCM-T/A-TC-FlAsH strategy. At low RIF pressure, we observed increased levels of both RelE and RelB, peaking at the minimum inhibitory concentration (MIC), showing a 5.4-fold and 1.3-fold enhancement, respectively, compared to the non-stressed condition (Fig. 2A-i, ii). Contrary to the common belief that antitoxins are selectively degraded under stress, our results revealed, for the first time, an increase in antitoxin RelB expression under low-dose RIF treatment. However, as the RIF concentration increased beyond 1 MIC, the levels of both RelE and RelB decreased dramatically. Although the level of RelE decreased compared to its peak at 1 MIC, it remained elevated relative to the non-stressed condition. In contrast, the RelB level significantly decreased at concentrations above 1 MIC. This distinct expression pattern of RelE and RelB, characterized by differential responses below and above 1 MIC RIF pressure, has not been observed previously.

By calculating the T/A ratio of RelE and RelB, we found that it increased by 3.0 to 8.5-fold under varying RIF concentrations (from 0.5 to 100 MIC) compared to the non-stressed condition (Fig. 2A-iii), indicating that E. coli responds to antibiotic stimuli by elevating the T/A ratio. Additionally, RelE and RelB exist in several forms: RelB$_2$, RelB$_2$RelE, RelB$_2$RelE$_2$, and RelE, each with different affinities for their operator–a phenomenon known as conditional cooperativity (Boggild et al, 2012; Overgaard et al, 2008). As shown in Fig. 2B, when the T/A ratio is less than 0.5 ([RelB$_2$] > [RelE]),

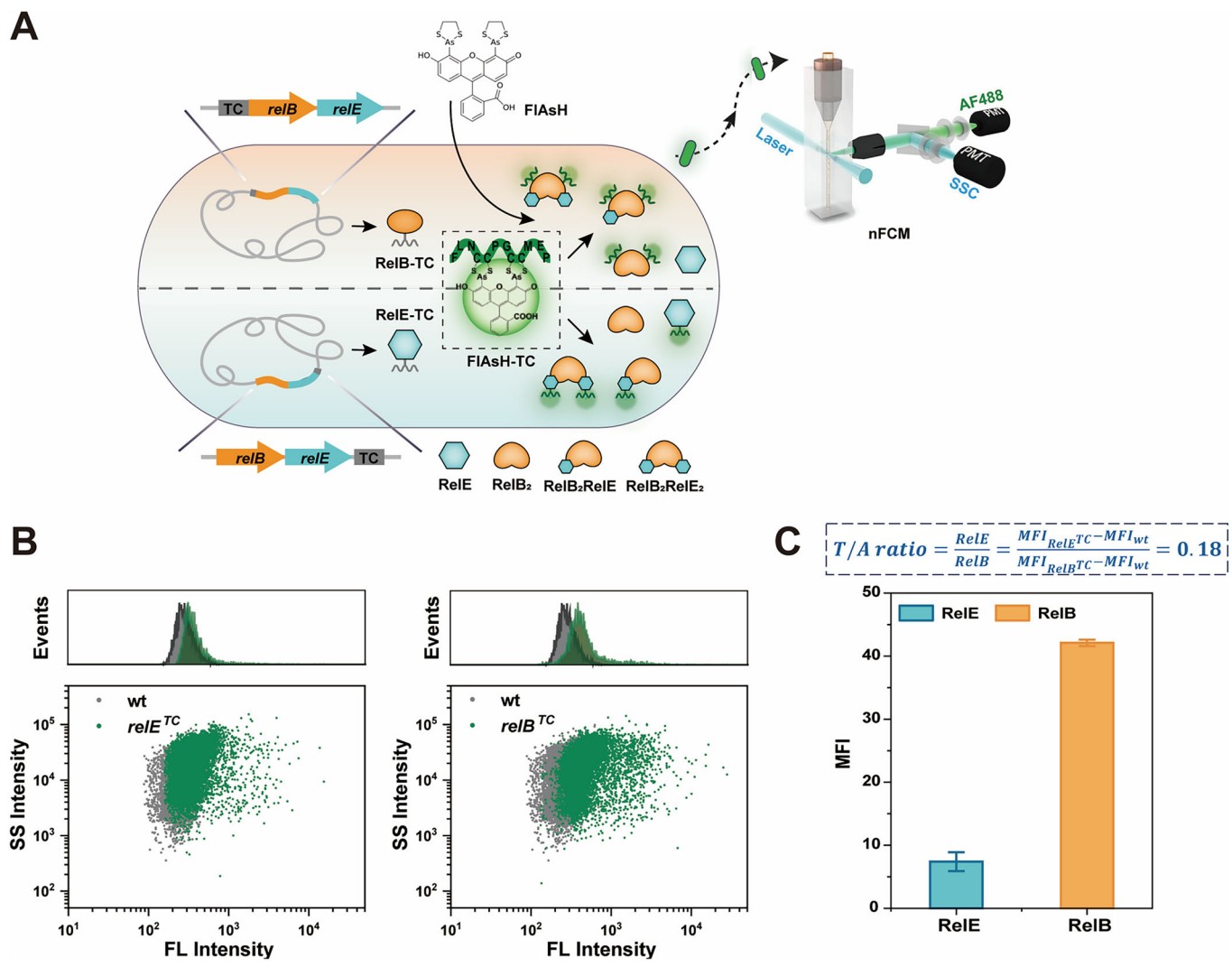

**Figure 1.   Quantification of TA expression at the single-cell level via the nFCM-T/A-TC-FlAsH strategy.**

(**A**) Schematic illustration of the construction of in situ TC-tagged *relE* and *relB* strains, and the TC-FlAsH labeling strategy for single-cell analysis of RelE and RelB using nFCM. (**B**) Bivariate dot plots of side scatter (SS) versus fluorescence (FL) and the corresponding FL histograms for the wild-type (wt), *relE*$^{TC}$ and *relB*$^{TC}$ strains during exponential growth. (**C**) Median fluorescence intensity (MFI) of *relE*$^{TC}$ and *relB*$^{TC}$ strains compared to the wild-type control, demonstrating the relative expression levels of RelE and RelB. Data in (**C**) are presented as mean ± SD, $n = 3$ individual experiments. Source data are available online for this figure.

RelB$_2$ and RelB$_2$RelE mainly exist in bacteria, binding cooperatively to the *relBE* promoter and repressing transcription. This explains the relatively lower levels of RelE and RelB under non-stressed conditions. When bacteria are challenged with ≤1 MIC RIF, T/A ratios between 0.5 to 1 ([RelB] > [RelE] > [RelB$_2$]) lead to the formation of RelB$_2$RelE$_2$, derepressing the promoter because RelB$_2$RelE$_2$ does not bind to the operato (Overgaard et al, 2008). This corresponds with the increased abundance of both RelE and RelB (Fig. 2A-i, ii). When bacteria are challenged with >1 MIC RIF, T/A ratios > 1 ([RelE] > [RelB]), indicate the availability of free RelE. As an endonuclease, RelE cleaves mRNA and inhibits translation (Pedersen et al, 2003; Takagi et al, 2005). Consequently, the levels of RelE and RelB is lower than under low RIF pressure, particularly with antitoxin RelB being significantly lower than under non-stressed conditions and low RIF pressure due to instability. These results further confirm that the nFCM-T/A-TC-

FlAsH strategy allows accurate quantification of TA abundance and directly reveals that TA expression is regulated by the T/A ratio. Additionally, it is first proposed here that bacteria respond to antibiotic pressure by increasing T/A ratios in two distinct modes.

To assess the generality of our results, we performed analogous experiments with two additional bacteriostatic antibiotics: chloramphenicol (Cam) and tetracycline (TET) (Fig. 2C,D). Similar to the results obtained with RIF, peak levels of RelE and RelB was observed at 1 MIC, and the T/A ratio was higher than in the non-stressed condition under all Cam and TET concentrations (Fig. 2C,D-i, ii). However, for Cam, within the concentration range of 0.5–100 MIC, the levels of both RelE and RelB was consistently higher than in the non-stressed condition. The T/A ratio analysis (Fig. 2C-iii) indicated that RelE/RelB was in the transcriptional activation range of 0.5 < T/A < 1.0, resulting in increased production of both RelE and RelB. For TET, when exposed to low pressure of 0.5–20 MIC, the content of RelE

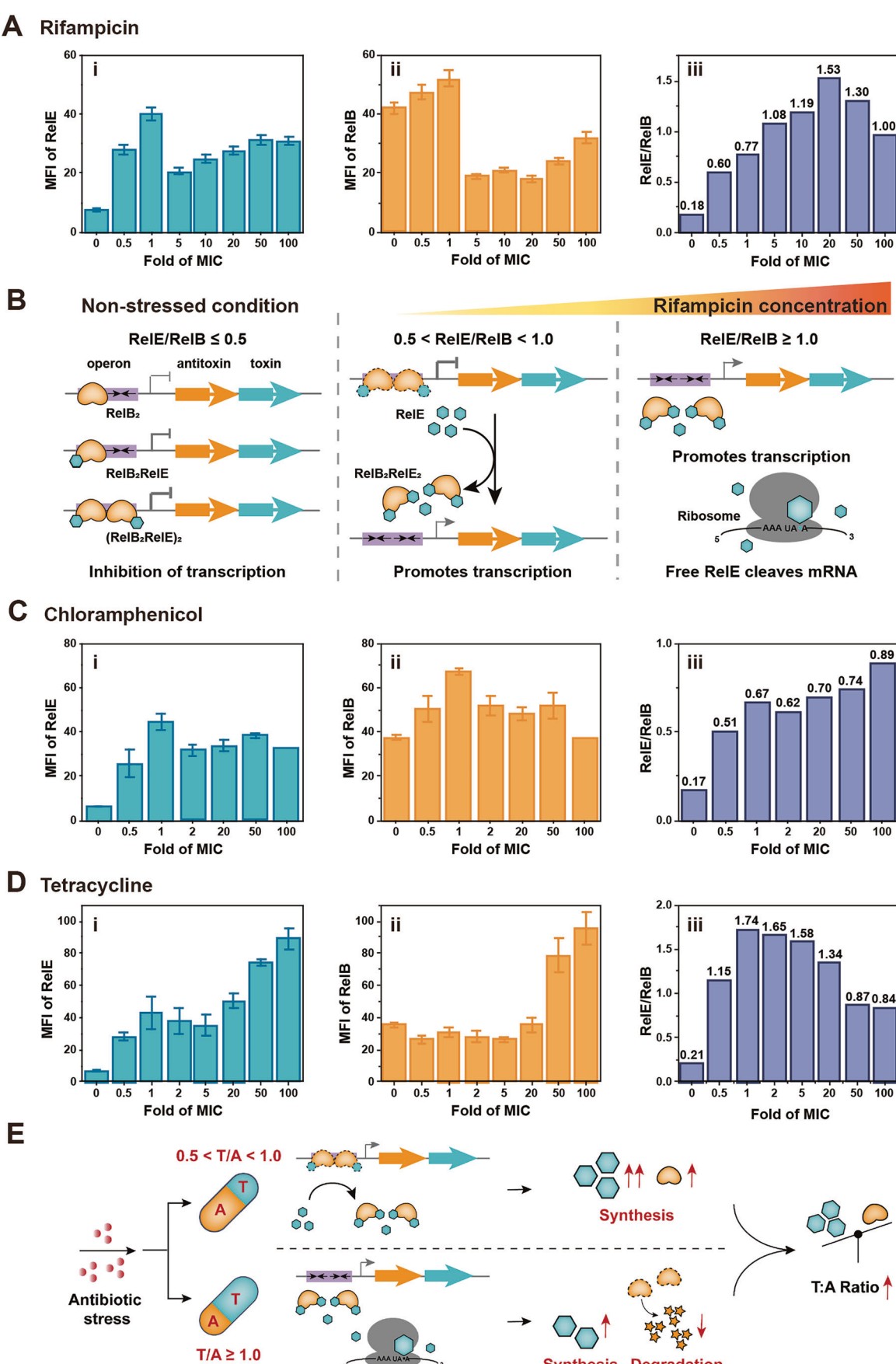

◄ **Figure 2. Quantification of RelE and RelB production in bacteriostatic antibiotic-treated bacteria using the nFCM-T/A-TC-FlAsH strategy.**

(A) The levels of RelE (i) and RelB (ii), and the corresponding T/A ratios (iii) in bacteria exposed for 2 hr to a series of RIF dosages (0, 0.5, 1, 5, 10, 20, 50, and 100 MIC; $MIC_{RIF} = 10$ µg/mL). (B) Model summarizing how conditional cooperativity mediated by T/A ratios regulates the differential expression of the TA system. (C, D) The levels of RelE (i) and RelB (ii), and T/A ratios (iii) in bacteria exposed to a series of Cam (C) and TET (D) dosages (0, 0.5, 1, 5, 10, 20, 50, and 100 MIC, $MIC_{Cam} = 2$ µg/mL; $MIC_{TET} = 0.5$ µg/mL). (E) Schematic illustration of how bacteria increase the T/A ratio through two distinct modes, with T/A = 1.0 serving as the boundary to cope with antibiotic pressure. Data in (A, C, D) are presented as mean ± SD, n = 3 individual experiments. Source data are available online for this figure.

increased while RelB decreased compared to non-stressed condition, consistent with T/A > 1.0 promoting transcription with free RelE available to cleave mRNA. At 50 and 100 MIC concentrations, both RelE and RelB increased considerably, consistent with 0.5 < T/A < 1.0. These results further indicate that the T/A ratio regulates the expression patterns of toxin and antitoxin. Using the nFCM-T/A-TC-FlAsH strategy, we found that bacteria increase the T/A ratio through two distinct modes in response to antibiotic pressure with T/A = 1 as the boundary: When 0.5 < T/A < 1.0, the levels of both the toxin and antitoxin increase, with the toxin level rising more significantly; When T/A ≥ 1.0, the level of RelE continues to increase, while that of RelB decreases (Fig. 2E).

## Positive linear correlation between bacterial persistence and T/A ratio

Mathematical models support the idea that the conditional cooperativity-mediated bistability underlies the type II TA system's ability to induce persister formation (Cataudella et al, 2013; Cataudella et al, 2012). Having demonstrated that bacteria increase the T/A ratio in response to antibiotic pressure, we investigated the linkage between this ratio and persistence. Quantitative persistence assays showed that pretreatment with RIF significantly improved bacterial survival under subsequent ampicillin (AMP) treatment (Fig. 3A), consistent with previous studies (Kwan et al, 2013). Bacterial survival increased with higher RIF concentrations, peaking at 96.2% at 50 MIC, before slightly dropping at 100 MIC. Standard disk-diffusion assay did not reveal any reduction in the inhibition zone compared to the non-stressed condition growth strain (Fig. 3B, first row), indicating that the increased survival was not due to alterations in resistance. Further, the Tolerance Disk Test (Gefen et al, 2017) (TD test), performed by replacing the antibiotic disk with a new disk impregnated with nutrients, showed detectable colonies forming in the inhibition zone. This confirmed that persistence, rather than resistance, was responsible for the improved survival rate (Fig. 3B, second row). The bacterial lawn observed in the inhibition zone indicates high persistence in the 1 MIC RIF-treated strain. For the whole population, the T/A ratio exhibited a strong linear correlation with survival ($R^2 = 0.9407$), suggesting that higher T/A ratios correspond to improved survival (Fig. 3C). Combined, our results convincingly demonstrate that bacteria cope with external antibiotic pressure by increasing the T/A ratio, which facilitates persistence.

## Single persister detection reveals a positive correlation between heterogeneous RelE levels and dormancy duration

To avoid the tiny fraction of persisters being overlooked in bulk population studies, we further examined the correlation between RelBE levels and persistence in single persister cells. Persisters from *E. coli* pretreated with 0.5, 1, 5, and 10 MIC RIF for 2 h were isolated by subsequent AMP treatment for 3 h; AMP lyses the

sensitive non-persister cells while leaving the persisters intact (Kwan et al, 2013). Figure 4A presents bivariate dot plots of side scatter (SS) versus RelE fluorescence and the fluorescence distribution histograms for individual persisters and non-stressed condition cultivated $relE^{TC}$ (negative control), respectively. Our results demonstrate significant variability in RelE production among individual persister cells, highlighting that single-cell data capture gradual protein level changes that bulk data might miss. Interestingly, although most persisters exhibited high levels of RelE, some individual cells showed low levels comparable to sensitive cells. This finding challenges the prevailing view that higher toxin expression is a hallmark of persisters. Furthermore, given that *E. coli* possesses multiple TA systems, persisters with lower RelE levels may compensate by elevated levels of other toxins. This highlights the complexity of the persistence phenomenon and suggests that multiple mechanisms may coexist within the persister population. Similarly, RelB production under the same conditions showed no significant variation between non-stressed and persister cells (Appendix Fig. S2), suggesting that RelB may not be the main factor mediating persistence in persisters. For comparison, the MFI of RelE, RelB and the RelE/RelB T/A ratio are plotted in Fig. 4B,C, respectively. RelE production was significantly increased at the persister level, peaking at 1 MIC, while the T/A ratios of 0.5 to 10 MIC RIF-pretreated persisters were all above 1, indicating higher toxin production compared to antitoxin in persisters. Interestingly, the T/A ratio was highest at 0.5 MIC and decreased with increasing RIF concentration, contrasting with the continuous increase observed for the whole population average.

To assess the dependence of bacterial persistence on RelE levels, we used fluorescence-activated cell sorting (FACS) to segregate the population into three subpopulations with low, median and high fluorescence ($RelE_{low}$, $RelE_{median}$ and $RelE_{high}$) after 1 MIC RIF treatment. These subpopulations were exposed to fresh media or AMP alongside total persister cells and non-stressed cells to measure the distribution of growth resumption and antibiotic susceptibility (Fig. 4D). As expected, non-stressed cells began proliferating earlier than persisters, both as a whole and within subpopulations, confirming that persisters exhibit delayed growth resumption (Fig. 4E,F). The proliferation rates of persisters decreased with increasing RelE level, with the overall persister proliferation rate falling between those of the $RelE_{high}$ and $RelE_{median}$ subpopulations. These results indicate that dormancy is a heterogeneous physiological state with varying depths, directly correlated with the RelE level within persisters. Lag times calculated from growth curves using the Gompertz equation further confirmed that higher RelE production contributes to a deeper dormancy state (Fig. 4F). Thus, high cellular concentrations of RelE significantly deepen dormancy, helping persister cells evade antibiotic killing and enhancing population survival. Furthermore, RelE production may be crucial not only for the induction of persisters but also for their resuscitation.

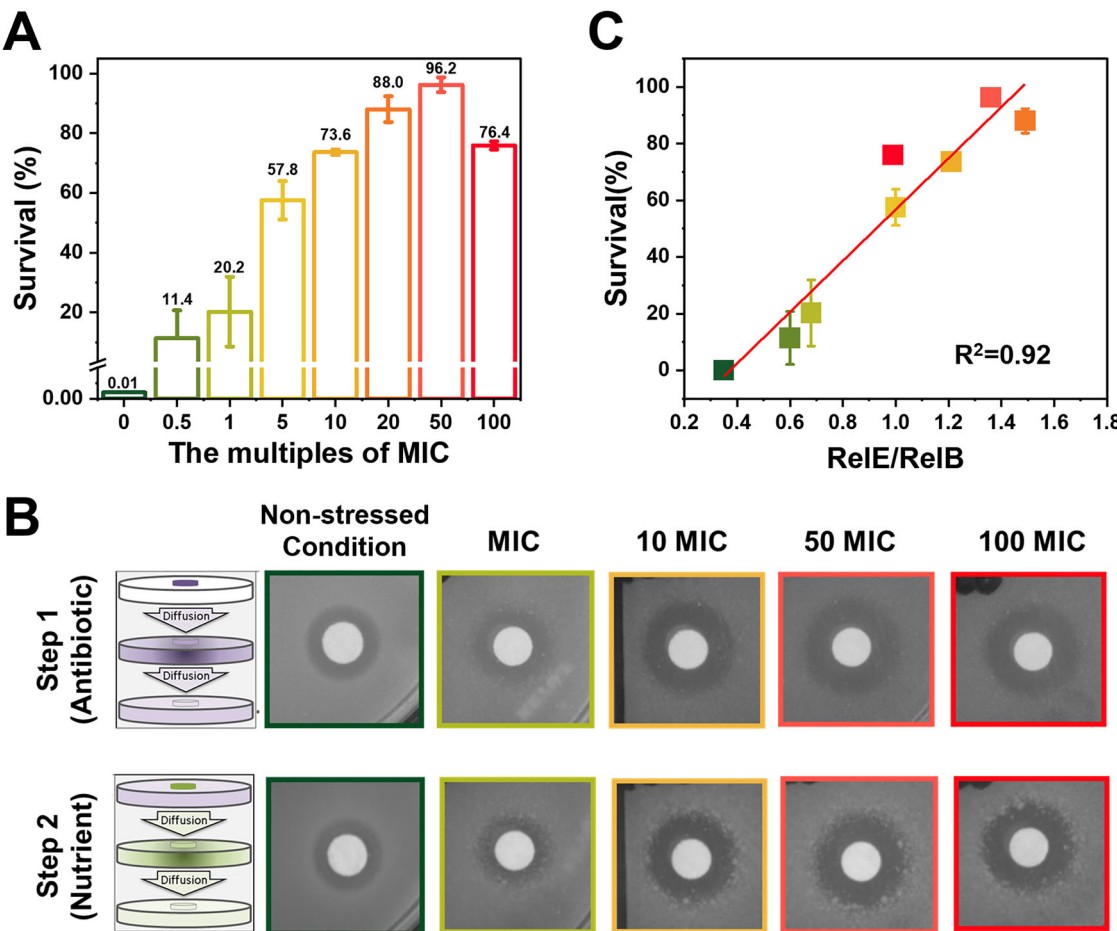

**Figure 3. Correlation analysis between bacterial persistence and T/A ratio under RIF stress using whole population averages.**

(A) Frequency of persister formation in bacteria pretreated with varying concentrations of RIF, determined by antibiotic (ampicillin) susceptibility measurements. (B) Disk-diffusion assays comparing strains treated with RIF to untreated controls. The first row shows standard disk-diffusion results, while the second row shows tolerance disk test (TD test) results indicating persistence alterations. (C) Linear correlation between the frequency of persister formation and the T/A ratio as a population average for RIF-treated bacteria. Data in (A, C) are presented as mean ± SD, $n = 2$ individual experiments. Source data are available online for this figure.

## Proteomic profiling of persisters with divergent RelE levels

To elucidate the distinct persistence phenotypes driven by different RelE levels, label-free quantitative proteomics analyses were conducted to investigate the proteomic changes among non-stressed, RelE$_{low}$, and RelE$_{high}$ groups. A total of 2183 proteins were obtained. Principal component analysis (PCA) data clearly demonstrated distinct protein expression profiles among the three groups, with both RelE$_{low}$ and RelE$_{high}$ groups being clearly separated from the Non-stressed group (Fig. 5A). Differential abundance proteins were identified based on abundance ratios greater than 2 or less than 0.5, corresponding to increased and decreased expression, respectively (Fig. 5B). A total of 506, 246, and 186 proteins exhibited altered abundance between the RelE$_{high}$ and non-stressed group, RelE$_{low}$ and non-stressed group, and RelE$_{high}$ and RelE$_{low}$ group, respectively, as visualized by volcano plots and Venn diagrams (Fig. 5C,D). Specifically, 285 upregulated and 221 downregulated proteins were detected in the RelE$_{high}$ group compared to the Non-stressed group, while 33 upregulated and

213 downregulated proteins were identified in the RelE$_{low}$ group compared to the non-stressed group. Additionally, 182 upregulated and 4 downregulated proteins were detected in the RelE$_{high}$ group compared to the RelE$_{low}$ group. Comparative analysis revealed 165 common differentially expressed proteins (DEPs) between the RelE$_{high}$ vs. Non-stressed group and RelE$_{low}$ vs. Non-stressed group (Fig. 5D). Subcellular localization analysis was performed to determine the functional sites of the DEPs. The results indicated that DEPs were distributed across five cellular locations: inner membrane, cytoplasm, secreted, periplasm, and outer membrane (Fig. 5E).

Gene Ontology (GO) analysis was used to visualize the biological processes, cellular components, and molecular functions influenced by high or low RelE production. A total of ten biological processes were significantly enriched (Fig. 5F-i). Notably, many of these processes were related to responses to stimuli, drugs and radiation, as well as antibiotic and drug transport, aligning with the findings that high RelE levels significantly enhance drug tolerance (Fig. 3). Correspondingly, DEPs were predominantly localized in membrane-associated parts (Fig. 5F-ii). Furthermore, molecular

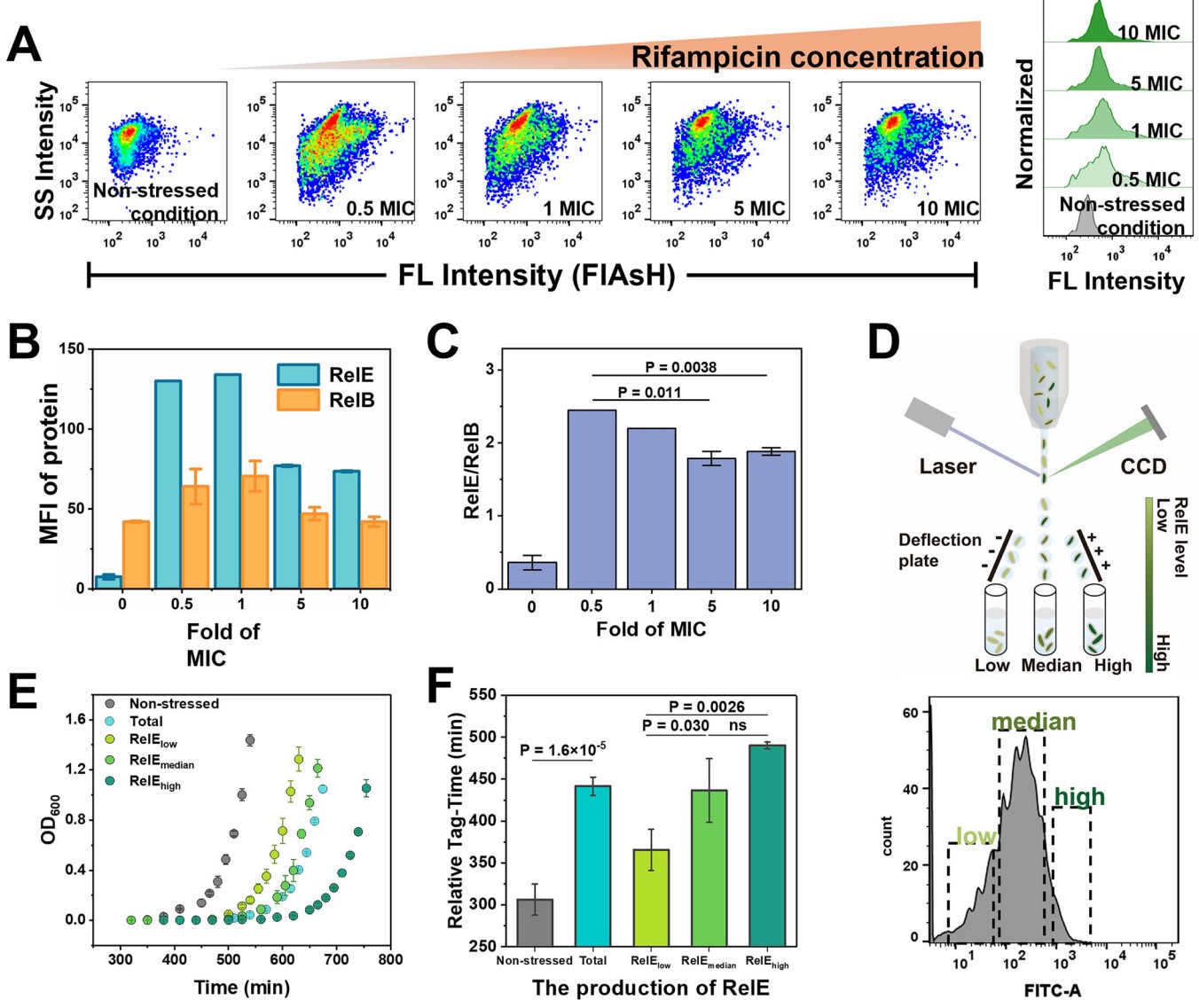

**Figure 4. Exploration of RelE production and its association with persistence in single persister cells.**

(A) Bivariate dot plots of SS vs. FL for RelE, along with the corresponding FL histograms for *relE^TC* persisters purified from cells pretreated with 0.5, 1, 5, and 10 MIC RIF, followed by AMP treatment, and total cells cultivated under non-stressed conditions as a negative control. (B, C) RelE and RelB levels (B) and T/A ratios (C) for the four persister conditions. (D) Schematic illustration of sorting persister cells based on RelE abundance levels. (E, F) Growth curve (E) and relative lag times (F) of the four groups sorted by FACS for 1 MIC RIF pretreatment. The bars indicate the mean of at least three independent experiments; error bars represent the SD. Data in (B, C, E, F) are presented as mean ± SD, $n = 3$ individual experiments. Exact *p* values below 0.05 from One-way ANOVA are indicated above the corresponding box when compared to the control samples with "ns" denoting non-significant. Source data are available online for this figure.

function analysis revealed that high RelE levels significantly affect the transmembrane transport system, with five out of the ten enriched molecular functions involving transmembrane transporter activity proteins (Fig. 5F-iii).

## nFCM-flWGA reveals heterogeneity in persister resuscitation, correlating negatively with RelE expression and T/A ratio

Our study has established that the T/A ratio is a robust indicator for entry into the persistent state (Figs. 3C and 4C). To further

elucidate the role of RelE, RelB, and the T/A ratio in persister cell resuscitation, we developed a cell wall growth-based fluorescence dilution method to monitor bacterial division at the single-cell level using flow cytometry. We labeled the *N*-acetylglucosamine subunits forming the backbone of *E. coli* BW25113 cell walls with wheat germ agglutinin conjugated to Alexa Flour 633 (flWGA) (Ursell et al, 2014), allowing us to track cell division by monitoring the decrease in fluorescence intensity per cell (Fig. 6A).To validate this approach, post-incubation and removal of excess flWGA, persister cells that were formed with 1 MIC RIF and AMP treatment were diluted to entry the exponential phase, and their fluorescence

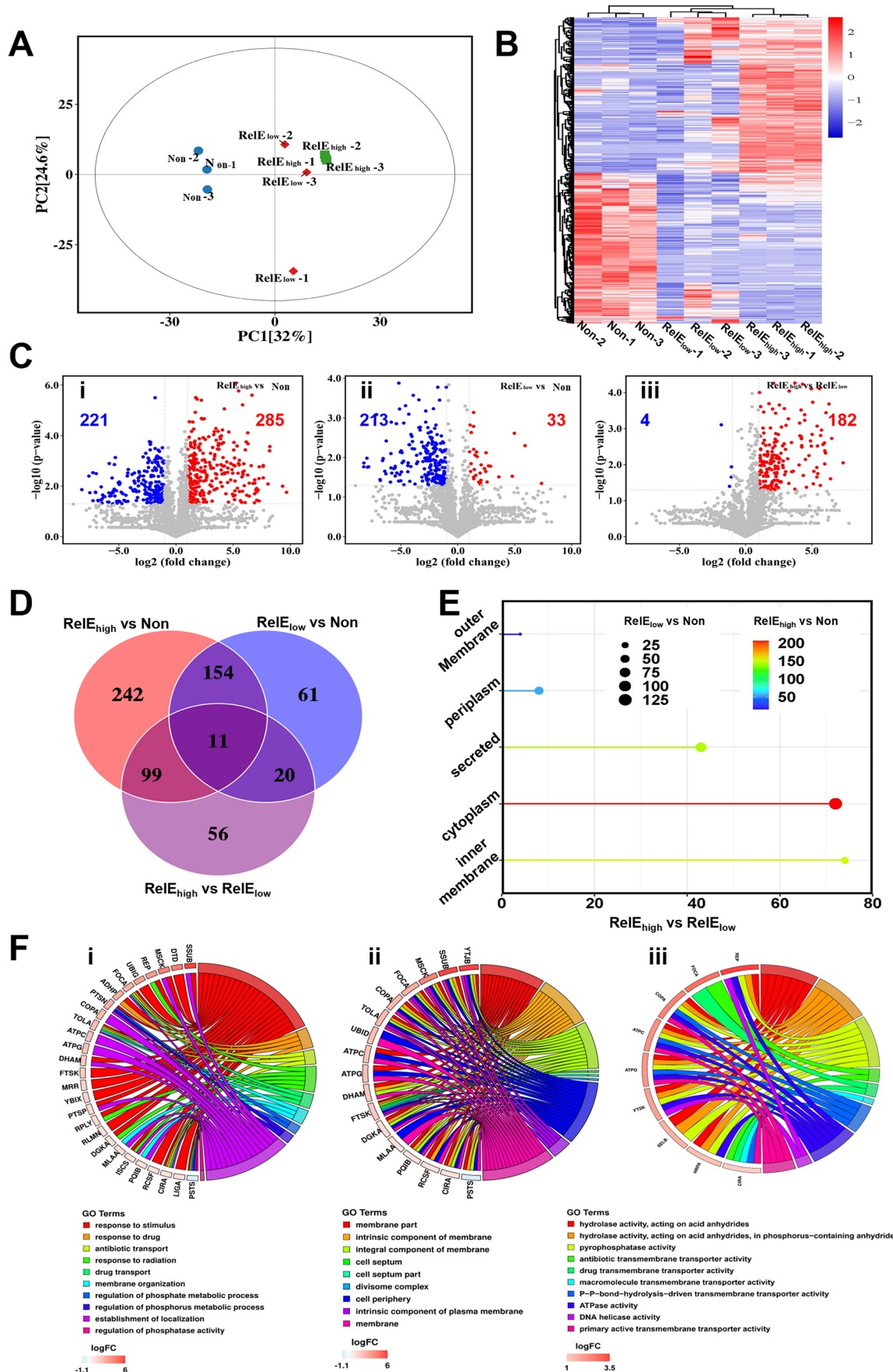

**Figure 5.   Proteomic analysis of persisters with low and high RelE production.**

(A) Principal component analysis (PCA) illustrating the separation between RelE$_{high}$, RelE$_{low}$ and non-stressed groups. Each dot represents technique replicates. (B) Hierarchical cluster analysis of 2173 proteins exhibiting statistically significant differences ($p < 0.05$; fold change >2 or <0.5) among RelE$_{high}$, RelE$_{low}$ and non-stressed cells. (C) Volcano plot displaying the p-values vs. log2 fold changes in protein abundance between RelE$_{high}$, RelE$_{low}$ and non-stressed cells. $p$ values were determined with Student's two-tailed $t$-test, $n = 3$ individual experiments. (D) Venn diagram generated using UpSetR, illustrating the overlap of differentially expressed proteins among the three comparisons. (E) Subcellular localization analysis of differentially expressed proteins. (F) Gene Ontology (GO) term enrichment analysis of differentially expressed proteins: Chord chart showing GO annotations for biological processes (i), cellular components (ii), and molecular functions (iii). Source data are available online for this figure.

signals were analyzed using nFCM (Fig. 6B). A homogeneous replication pattern was observed within the bacterial population, with the fluorescence signal gradually declining over time as confirmed by microscopy (Fig. 6C). The MFI decreased from 5985 to 17 within 3 h (Fig. 6D), correlating with the proliferation rate based on increasing bacterial concentration (Fig. 6E). The fluorescence intensity reduction per proliferative generation was calculated to be 1.995, indicating the effectiveness of the nFCM-flWGA fluorescence dilution method for measuring bacterial replication (Fig. 6F). Unlike traditional methods relying on chromosomally integrated inducible fluorescence proteins (e.g., GFP (Peyrusson et al, 2020) or mCherry (Orman and Brynildsen, 2015)), our cell wall growth-based approach does not require genetic modification of bacteria, making it appropriate for direct use on clinical antibiotic tolerance samples.

Next, we applied the nFCM-flWGA method to monitor the resuscitation of persisters pretreated with various RIF concentrations at the single-cell level after antibiotic removal. Owing to the superior sensitivity of the nFCM, non-growing and dividing cells are expected to exhibit different features of SS and FL signals. By simultaneously measuring SS and FL, we distinguished dividing bacteria (higher SS and lower FL, region I) from non-growing cells (lower SS and higher FL, region II) and background signals (region III) (Fig. 6F). From the dot plots of SS versus the FL of WGA (Fig. 6F), we observed the proportion of dormant cells in region II gradually decreased over time as they transitioned to region I, with significant heterogeneity in proliferation rate among persisters. Interestingly, low doses of RIF resulted in slower recovery of proliferative capacity. Persisters pretreated with 5 and 10 MIC RIF began resuscitating within one hour, with 27 and 46% of cells falling in region I, while those pretreated with 0.5 and 1 MIC RIF showed deeper dormancy, initiating resuscitation at two and three hours, respectively. After 3 h, the remaining dormant cells were 80, 53, 7, and 4% for persisters pretreated with 0.5, 1, 5, and 10 MIC RIF, respectively. Given that antibiotic persistence promotes resistance evolution (Levin-Reisman et al, 2017), the deeper dormancy observed at lower antibiotic doses may contribute to rising rates of antibiotic resistance. These findings underscore the importance of maximizing antibiotic exposure in clinical treatments, as indicated by pharmacokinetic and pharmaco-dynamic models (Roberts et al, 2008).

Further analysis revealed a positive linear correlation between RelE levels and the percentage of non-growing cells at all three resuscitation times (Fig. 6G), demonstrating a strong direct correlation between RelE concentration and dormancy duration. Additionally, the correlation between the T/A ratio and the percentage of non-growing cells increased over time ($R^2$ from 0.5820 at 1 h to 0.9770 at 3 h) (Fig. 6H), with the low correlation at 1 h attributable to the completely dormant state of both 0.5 and 1 MIC samples. These findings suggest that toxin levels dominate the

T/A ratio in persister cells and play a crucial role in the resuscitation process.

## Single-cell analysis reveals RelE reduction as a trigger for persister resuscitation through a low T/A ratio

The resurgence of infection is often linked to the resuscitation and regrowth of persisters under favorable conditions. Deciphering the mechanisms that extricate persisters from dormancy is therefore pivotal. We harnessed the nFCM-T/A-TC-FlAsH technique to investigate the dynamic production of RelE at a single-cell level during the resuscitation and proliferation of persisters. Figure 7A presents the bivariate dot-plot depicting SS versus RelE levels for persisters pretreated with 1 MIC RIF resuscitating from 0 to 2 h. We observed marked heterogeneity in RelE production throughout recovery. To elucidate these real-time dynamics, we categorized cells into three distinct populations: low RelE level and low SS (T$_{low}$), median RelE level and high SS (T$_{median}$), and high RelE level and median SS (T$_{high}$). Within the first 10 min (Fig. 7A-ii), the proportions of bacteria in T$_{median}$ and T$_{high}$ decreased from 37.0% and 27.5% to 31.9% and 17.3%, respectively. All of the 15.3% bacteria were subtracted from these two populations relocating to T$_{low}$ (rising from 35.5% to 50.8%). In the subsequent 10–20 min (Fig. 7A-iii), cells continued migrating from T$_{median}$ to T$_{low}$, with proportions adjusting from 31.9 to 25.7%, while T$_{high}$ population remained dormant, underscoring a non-homogeneous awaking process. By 30 min, the proportion of T$_{median}$ began climbing, reaching 73.4% at 120 min, as T$_{low}$ and T$_{high}$ proportions dwindled to 18.2 and 8.4%, respectively. Figure 7B illustrates the changes in MFI for T$_{low}$ and T$_{median}$ over the resuscitation timeline. Notably, RelE production in T$_{low}$ precipitously declined during the initial 30 min, with MFI plummeting from 27.2 to 0.2, indicating aggressive RelE degradation of RelE in all the three regions when persisters withdrew from pressure. From 60 to 120 min, as the proportion of bacteria in T$_{median}$ increased significantly, the fluorescence in this region rose from 87.6 to 120.6 (Fig. 7B). We hypothesize that severe RelE depletion is critical for initiating bacterial proliferation. Optical density (OD) measurements con-firmed this hypothesis as OD values remained stable prior to 30 min, rising discernibly by 60 min (Fig. 7C), corroborating bacterial growth. Consistent with these findings, comprehensive analyses of RelE and RelB levels revealed that RelE reached its nadir at 60 min (Fig. 7D), while RelB levels exhibited minimal fluctuation during the culture period (Fig. 7E). This culminated in a continuous decline in the T/A ratio, which reached a trough of 0.84 at 60 min (Fig. 7F), synchronizing with the initiation of bacterial proliferation (Fig. 7C). We propose that the reduction of RelE—and the consequent decrease in the T/A ratio—serves as a pivotal mechanism facilitating persister cell regrowth, potentially

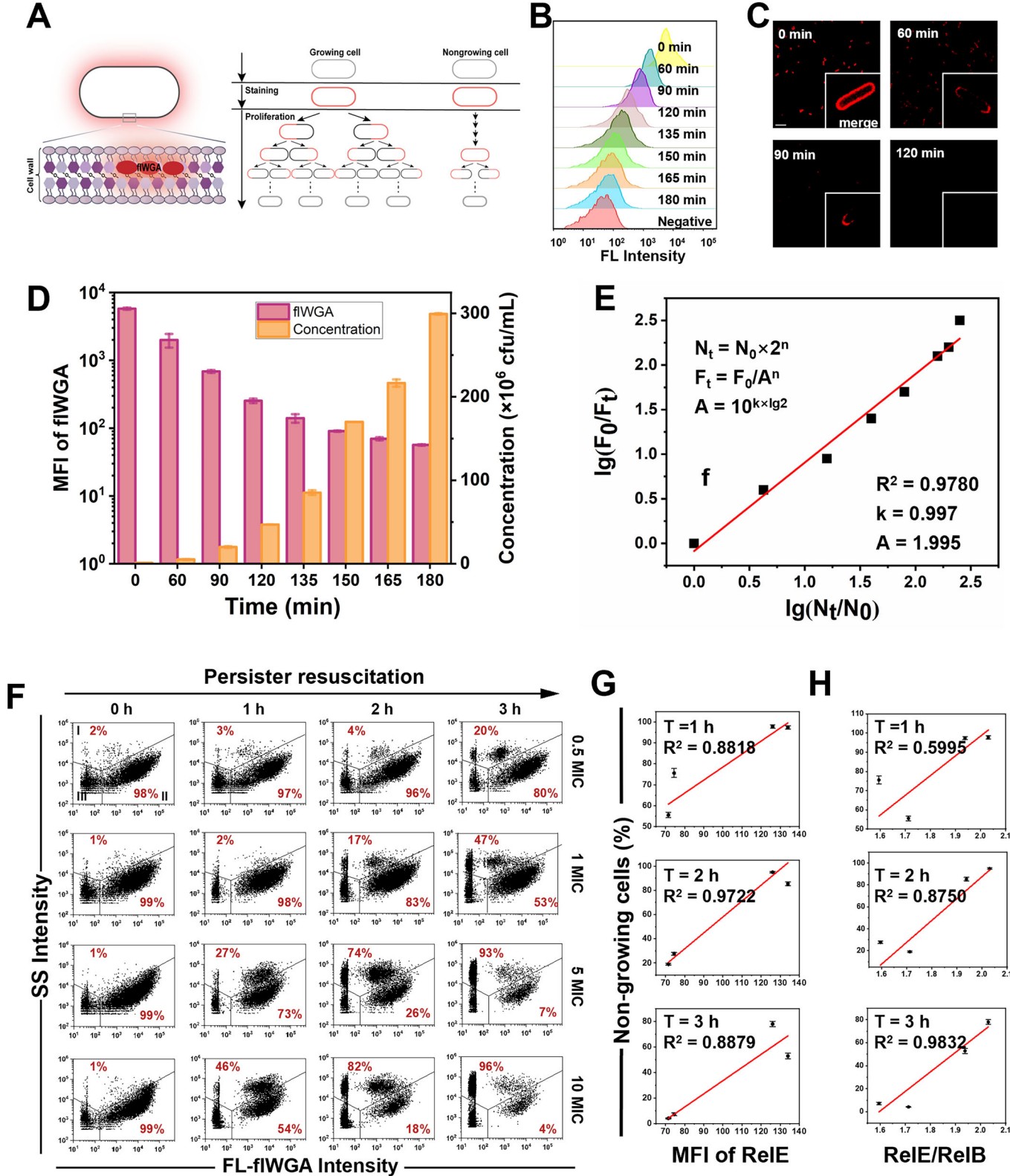

establishing a threshold for proliferation onset. Intriguingly, after 120 min of resuscitation, RelE production increased, and the T/A ratio increased above 1 again, suggesting the bacteria had not yet fully exited the persister state.

We further simultaneously monitored the dynamics of RelE production and the recovery state of persisters over an extended period. As illustrated in Fig. 7G, flWGA-stained 1 MIC RIF-pretreated persisters were cultured in Luria-Bertani (LB) medium. At specified

**Figure 6. Development and characterization of the nFCM-flWGA strategy for single-cell monitoring of persister resuscitation dynamics.**

(A) Schematic diagram illustrating bacterial proliferation monitoring at the single-cell level using the flWGA fluorescence dilution-based method. (B) Histograms of the fluorescence intensity distribution for bacterial proliferation over time after flWGA labeling, measured by flow cytometry. (C) Corresponding confocal images (scale bar: 2 μm). (D) MFI of flWGA and bacterial concentration over time. (E) Relationship between $lg(F_O/F_t)$ and $lg(N_t/N_O)$, indicating that the fluorescence intensity of single bacteria decreases by half in each proliferative generation. The number of cells at time $t$ ($N_t$) is calculated as $N_t = N_O \times 2^n$, where $N_O$ denotes the initial cell number and n denotes the generation of multiplication. For the single bacterial fluorescence signal at time t ($F_t$), $F_t = \frac{F_O}{A^n}$, where $F_O$ represents the initial fluorescence signal intensity, and

1/A denotes the percentage decrease in fluorescence signal intensity per generation in individual bacteria. A can be obtained as $A = 10^{lg2 \times \frac{lg\frac{F_O}{F_t}}{lg\frac{N_t}{N_O}}}$. By plotting $lg(F_O/F_t)$ on the y-axis and $lg(N_t/N_O)$ on the x-axis, a linear fit with slope k = 0.997 is obtained, simplifying A to $A = 10^{lg2 \times k}$. (F) Bivariate dot plots of SS versus FL-flWGA for the four persister conditions during resuscitation over time. (G, H) Correlation between RelE levels (G), T/A ratios (H), and the percent of non-growing cells. Data in (D, G, H) are presented as mean ± SD, $n = 3$ individual experiments. Source data are available online for this figure.

time points, aliquots were split into two: one for assessing resuscitation state via nFCM-flWGA, and the other for RelE production analysis using the nFCM-T/A-TC-FlAsH strategy. Correlated bivariate dot plots of SS versus flWGA and RelE production are shown in Fig. 7H; Appendix Fig. S3A. During the initial 30 min, bacteria transitioned from $T_{median}$ and $T_{high}$ to $T_{low}$ (Fig. 7A; Appendix Fig. S3A), while the proportion of dormant bacteria (in region I) remained relatively constant (Fig. 7H), suggesting that toxin degradation precedes cellular proliferation during resuscitation. At 60 min, cells in region II of Fig. 7H-iv increased to 10.8%, corroborating a slight rise in OD measurements. At 2 h, only 22.2% of cells were categorized in region II (Fig. 7H-v), implying that the majority remained in a persister state. Over the subsequent three hours, a sequential resuscitation and proliferation of persisters was observed (Fig. 7H vi-viii), accompanied by a gradual increase in RelE MFI, reaching its apex at 4 h (Appendix Fig. S3B). These data visually depict how the nadir of RelE production at 60 min acts as a trigger for initiating persister proliferation, with restored protein synthesis subsequently elevating RelE levels. By 5 h, nearly all bacterial cells (96.3%) had proliferated (Fig. 7H-viii), and the RelE MFI decreased substantially (Appendix Fig. S3B). To determine whether persisters had fully recovered to a sensitive state, we compared RelE production between non-stressed control bacteria and persisters at 0 and 5 h. As shown in Appendix Fig. S3C, by 5 h, RelE levels in both populations aligned, with similar bivariate dot plots and fluorescence histograms. The T/A ratio was calculated to be 0.128, indicating a full recovery to normal levels by 5 h. In conclusion, our findings delineate a three-stage regulatory mechanism of persister resuscitation driven by RelE production. Initially, the heterogeneous downregulation of toxin production to minimal levels permits the initiation of bacterial proliferation. As proliferation advances, there is a resurgence in toxin production due to restored protein synthesis. Finally, a decline in toxin levels allows bacteria to resume normal growth (Fig. 7I).

## Discussion

In this study, we developed the nFCM-T/A-TC-FlAsH and nFCM-flWGA methodologies, providing a robust platform for high-throughput, single-cell analysis of TA module expression and bacterial persister states in their native context. Our findings highlight the critical role of the T/A ratio in both the induction and resuscitation of persisters. We elucidate that bacterial cells adapt to antibiotic stress by elevating the T/A ratio via two distinct expression modes, with a critical threshold set at T/A = 1.0. When 0.5 < T/A < 1.0, both toxin and antitoxin levels are elevated, although toxin production is notably more pronounced.

Conversely, a T/A ratio equal to or exceeding 1.0 is characterized by increased RelE levels coupled with a reduction in RelB. Unlike the concurrent modulation of toxin and antitoxin levels that occurs as *E. coli* enters the persister state, the reduction of the T/A ratio during state exit is primarily driven by decreased toxin levels. Specifically, the resuscitation and subsequent bacterial proliferation are triggered when the T/A ratio reaches a nadir.

Our single-cell analysis using nFCM-flWGA uncovers considerable heterogeneity in persister resuscitation, aligning with distinct, sequential stages of RelE reduction. This suggests that multiple intermediate steps are involved in the persister awakening process. The observed positive correlation between RelE levels and dormancy duration further substantiates the hypothesis that RelE destabilization is a critical molecular event in initiating persister reactivation.

Proteomic analysis complements these observations, revealing significant differences between persisters with high versus low RelE production, especially in membrane-associated proteins involved in stimulus response and drug efflux. Furthermore, the finding that lower antibiotic doses tend to result in deeper dormancy manifested through higher T/A ratios, alongside evidence supporting persistence as a facilitator of resistance evolution, underscores the necessity of optimizing antibiotic exposure strategies. These insights into the molecular mechanisms underlying the TA module's role in persister formation and awakening deepen our understanding of bacterial persistence. They also provide promising avenues for the development of targeted therapeutic interventions and explore innovative applications of TA modules in health and technology sectors.

It is crucial to recognize that the persister model characterized in this study was induced by pretreatment with bacteriostatic antibiotics. This methodological choice provides substantial advantages for experimental establishment and captures a clinically relevant state of antibiotic-induced persistence. Nonetheless, we acknowledge that the broader persister cell field includes cells arising from diverse triggers, such as direct exposure to bactericidal antibiotics. Thus, exploring the applicability of our platform to these naturally occurring persister populations without prior induction remains a vital future research direction. We foresee that the foundational framework established in this study will be adaptable to analyze these systems, potentially by integrating enrichment strategies, to enable comparative analyses of persister states across a comprehensive spectrum. Future research should focus on advancing methods for the simultaneous detection of toxin-antitoxin modules at single-cell resolution. Specifically, coupling TC tags with additional fluorophores could facilitate dual fluorescent labeling, allowing for the direct measurement of single-cell T/A ratios.

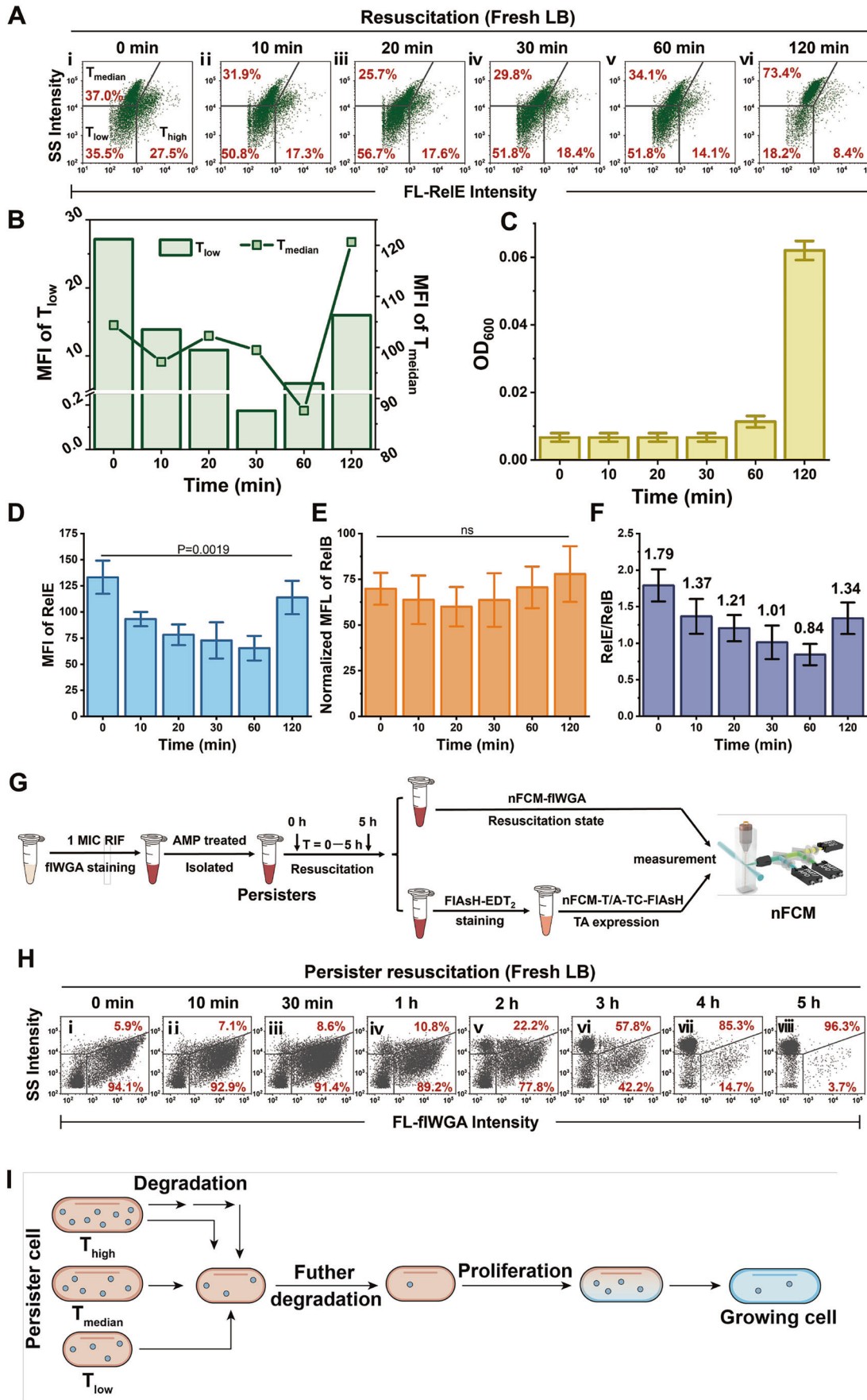

**A** Resuscitation (Fresh LB)

**G**

**H** Persister resuscitation (Fresh LB)

**I**

◄ **Figure 7. Single-cell analysis of RelE dynamics and persistence during resuscitation via nFCM.**

(**A**) Bivariate dot plots of SS versus FL for RelE in 1 MIC RIF-pretreated persisters during resuscitation over time. (**B**) Variation of normalized MFI of RelE in $T_{low}$ and $T_{median}$ regions. (**C**) Corresponding Optical density (OD) values over time. (**D–F**) MFIs of RelE (**D**), RelB (**E**), and the T/A ratio (**F**) during the resuscitation process. (**G**) Schematic diagram of the experimental procedure for simultaneous monitoring of persister resuscitation state and RelE production. (**H**) Bivariate dot plots of SS versus FL-flWGA, showing the resuscitation process of 1 MIC RIF pretreated persisters resuscitation over time. (**I**) Model summarizing the degradation of RelE in persisters at different rates to a low level, triggering bacterial proliferation. Data in (**C–F**) are presented as mean ± SD, $n = 3$ individual experiments. Exact $p$ values below 0.05 from one-way ANOVA are indicated above the corresponding box when compared to the control samples with "ns" denoting non-significant. Source data are available online for this figure.

# Methods

### Reagents and tools table

| Reagent/resource | Reference or Source | Identifier or Catalog Number |
|---|---|---|
| **Experimental models** | | |
| *Escherichia coli* BW25113 | Coli Genetic Stock Center, CGSC | Cat# 7636 |
| *Escherichia coli* BW25113 *relE*$^{TC}$ | This work | N/A |
| *Escherichia coli* BW25113 *relB*$^{TC}$ | This work | N/A |
| **Recombinant DNA** | | |
| pKD46 | Laboratory storage | N/A |
| pDNR-LIB | Laboratory storage | N/A |
| **Oligonucleotides and other sequence-based reagents** | | |
| agcagaagtgtcaccttcggtgcgaaacagagatgtcatgctttggttcaGAGTCCAAGCGAGCTCGATATCA | CmsacB-F, for pDNR-LIB cloning, this study | - |
| tagtgcgatacttgtaatgacatttgtaattacaagaggtgtaagacatgCACATATACCTGCCGTTCACTATTATTTAGTG | CmsacB-R, for pDNR-LIB cloning, this study | - |
| CAGATGACGATCAGGGCGATTAACATC | CmsacB-out-F, sequencing primer for pKD46 with cm-sacB, this study | - |
| GATCACCGTTCTTACGACTACTTTCTGAC | CmsacB-out-R, sequencing primer for pKD46 with cm-sacB, this study | - |
| agcagaagtgtcaccttcggtgcgaaacagagatgtcatgctttggttca | RelE-in-F, for relETC/relBTC homologous recombination, this study | - |
| GATCACCGTTCTTACGACTACTTTCTGAC | RelE-in-R, for relETC homologous recombination, this study | - |
| CAGATGACGATCAGGGCGATTAACATC | RelE-out-F, sequencing primer for relETC/relBTC, this study | - |
| GATCACCGTTCTTACGACTACTTTCTGAC | RelE-out-R, sequencing primer for relETC/relBTC, this study | - |
| catgtcttacacctcttgtaattacaaatgtcattacaagtatcgcacta | RelB-in-R, for relBTC homologous recombination, this study | - |
| **Chemicals, enzymes and other reagents** | | |
| Propidium iodide fluorescent dye | Invitrogen | Cat# P1304MP |
| Lysozyme | Sigma | Cat# L6876 |
| RNase | Thermo Fisher | Cat# EN0531 |
| Alexa Fluor 633-WGA | Invitrogen | Cat# W21404 |
| SYTO 9 | Thermo Fisher | Cat# S34854 |

| Reagent/resource | Reference or Source | Identifier or Catalog Number |
|---|---|---|
| Rifampicin | Sangon Biotech | Cat# A600812 |
| Tetracycline | Sangon Biotech | Cat# A600280 |
| Chloramphenicol | Sangon Biotech | Cat# A600118 |
| Ampicillin | Sangon Biotech | Cat# A100339 |
| **Software** | | |
| FlowJo 10.6.2 | FlowJo LLC, Ashland, OR, USA | https://www.flowjo.com/ |
| LabVIEW 8.5 | EMERSON, Saint. Louis, USA | https://www.ni.com/zh-cn/shop/labview.html |
| LAS X 3.7.3.23245 | Leica Microsystems GmbH, Wetzlar, Germany | https://www.leica-microsystems.com/ |
| BD FACSuite | Becton, Dickinson and Company, Franklin Lakes, NJ, USA | https://www.bdbiosciences.com/en-us |
| Adobe Photoshop | Adobe Inc, San Jose, CA, USA | https://www.adobe.com/ |
| ImageJ 1.52i | National institutes of Health, USA | https://imagej.nih.gov/ij/ |
| OriginPro 2017C | Northampton, MA, USA | https://www.originlab.com/ |
| **Other** | | |
| TC-FlAsH fluorescent dye kit | Invitrogen | Cat# T34561 |
| Genomic DNA purification kit | Tiangen | Cat# DP302 |
| Ligation sequencing kit | Oxford Nanopore | Cat# SQK-LSK109 |
| Native barcoding kit | Oxford Nanopore | Cat# EXP-NBD104 |

## Bacterial strains construction

The bacterial cell strains used in this study were derived from the *E. coli* BW25113, obtained from Coli Genetic Stock Center (CGSC). To construct the *relE*^TC^ strain, a two-step seamless recombineering method was employed using the *cm-sacB* selection cassette for precise genetic modifications (Sharan et al, 2009; Trokter and Waksman, 2018). In the first step, the *cm-sacB* cassette was inserted at the site immediately downstream of *relE* gene. This cassette was then replaced with the *tc* sequence in the second step. Briefly, the *cm-sacB* cassette was PCR amplified from pDNR-LIB using primers CmsacB-F and CmsacB-R, each containing a 50-bp sequence homologous to the *relE* region. The PCR products were electroporated into *E. coli* BW25113 cells harboring pKD46, which encodes λRed proteins. Chloramphenicol-resistant colonies were verified using colony PCR with primers CmsacB-out-F and CmsacB-out-R, followed by direct sequencing. Subsequently, the *cm-sacB* cassette at the native locus was replaced with the *tc* sequence. One colony was picked, incubated overnight at 30 °C and then plated on Luria-Bertani (LB) plates with 10% sucrose to select for plasmid excision from the chromosome by a second crossover. Verification was performed using colony PCR with primers RelE-in-F and RelE-in-R. The final mutant, *relE*^TC^, was confirmed by PCR amplification with primers RelE-out-F and RelE-out-R,

followed by direct sequencing. The *relB*^TC^ strain was constructed using the same protocol as *relE*^TC^, with the *tc* sequence inserted immediately upstream of the *relB* gene.

## Bacterial cell culturing

*E. coli* BW25113, *relE*^TC^, and *relB*^TC^ strains were grown in LB medium (10 g of tryptone, 5 g of yeast extract, and 10 g of NaCl per liter) or M9 minimal medium (12.8 g of $Na_2HPO_4 \cdot 7H_2O$, 3.0 g of $KH_2PO_4$, 0.5 g of NaCl, 1.0 g of $NH_4Cl$, 2 mL of 1 M $MgSO_4$, 0.1 mL of 1 M $CaCl_2$, 0.4% D-glucose per liter). Cultures were incubated in the exponential growth phase at 37 °C in baffled flasks on a 250 rpm rotary shaker.

## FlAsH labeling

When the bacterial cultures reached the exponential phase ($OD_{600}$ of 0.4), 30 μL of bacterial solution was preloaded with 5 μM $FlAsH\text{-}EDT_2$ and incubated at 37 °C in the dark for 1 h with shaking. The mixture was centrifuged (8000 × g, 8 min, 4 °C) and washed twice with 50 μL Hanks' balanced salt solution (1 × HBSS) containing 0.5 mM 2, 3-dimercaptopropanol (BAL). The bacterial cells were suspended in 20 μL PBS for nFCM analysis.

## WGA cell wall labeling

Cells were labeled with the N-acetylglucosamine-and sialic acid-specific lectin wheat germ agglutinin, conjugated to Alexa Fluor 633 (flWGA). Briefly, $relE^{TC}$ cells were grown to the exponential phase in LB, and $1.5 \times 10^8$ cfu E. coli were centrifugated and resuspended with 113 μL flWGA at final concentration of 1 mg/mL. The labeling ratio of flWGA was $1.33 \times 10^6$ cfu/μL. After incubating for 2 h with shaking (250 rpm) at 37 °C in the dark, the cells were washed twice with PBS to remove excess lectin. The bacterial cells were suspended in 20 μL PBS for nFCM and fluorescence microscopy analysis.

## nFCM measurement

The laboratory-built nFCM, equipped with three photomultiplier tubes (PMTs), was used for the simultaneous detection of side scatter (SS) and two fluorescence (FL) channels. A solid-state 488 nm continuous-wave laser (Newport) was used as the excitation source, with a laser excitation power of 4.5 mW. Bandpass filters of FF0-520/35 and FF0-700/40 were used for green fluorescence and red fluorescence detection, respectively. Data acquisition and processing were performed using a LabVIEW 8.5 software (National Instruments) program, with a data acquisition time of 3 min per sample. The 200 nm yellow-green FluoSpheres (Invitrogen) were used to adjust the nFCM to ensure consistent signals for both SS and fluorescence before each of the tests, which were also used to standardize the sample signals.

## Fluorescence microscopy analysis

Cells were imaged using a Leica SP8 confocal microscope with a 100× oil-immersion objective. The flWGA was excited with the 633-nm line of an argon laser. Fluorescence emissions were collected between 650 and 700 nm. Fluorescence channels were scanned sequentially, and all images were obtained under the same conditions. Images were processed and analyzed using LAS X 3.7.3.23245 and Adobe Photoshop.

## Cell sorting

Cell sorting was performed on a BD FCSAria III flow cytometer. Microorganisms were identified by forward scatter (FS) and side scatter (SS) parameters. FlAsH-labeled strains were sorted into three groups based on their fluorescence intensity (488-nm excitation with 530/30-nm bandpass filter) using a 70 μm nozzle. Approximately 10,000,000 cells were collected in each group.

## Experiments in broth

For the fluorescence dilution experiment, flWGA-labeled cells were diluted in fresh LB medium to reach a starting $OD_{600\ nm}$ of 0.005. Cultures were incubated at 37 °C, and aliquots were collected over time, washed, and suspended in PBS for flow cytometry or epifluorescence microscopy analysis. Growth curves were determined by $OD_{600}$ measurements, and the lag phase was calculated using the Gompertz equation. For time-kill curves, samples were diluted to a starting inoculum of $1 \times 10^5$ cfu/mL and exposed to 200 μg/mL ampicillin for the indicated times. For colony-forming

unit (cfu) counting, samples were diluted in PBS before plating on tryptic soy agar. Data were expressed as lg cfu/mL after the incubation period compared to the starting inoculum.

## Antibiotics response

$relE^{TC}$ or $relB^{TC}$ strains were grown to mid-exponential phase ($OD_{600\ nm} = 0.4$) in M9 minimal medium and exposed to Rifampicin (RIF), Chloramphenicol (Cam) or Tetracycline (TET) at a series of concentrations (0, 0.5, 1, 2, 5, 10, 20, 50, and 100 MIC) for 2 h with shaking (250 rpm). E. coli BW25113 cultivated without the stressor at 37 °C (non-stressed condition) served as a blank control to represent the autofluorescence background. Samples were extensively washed with PBS to reduce drug carryover and adjusted to ~$5 \times 10^8$ cfu/mL for FlAsH labeling and flow cytometry analysis.

## Antibiotic killing and persister isolation

Mid-exponential phase strains treated with various concentrations of RIF were washed and resuspended in fresh M9 medium supplemented with 200 μg/mL ampicillin (AMP) at $10^8$ cfu/mL, with shaking (250 rpm) at 37 °C for 3 h. Cells were then collected by centrifuging (8000 × g, 4 °C, 8 min) and washed with M9 minimal medium with 200 μg/mL AMP three times to isolate unlysed persister cells. Biphasic killing curves were performed to confirm that the persisters were from the second killing phase.

## Tolerance detection test (TD test)

The TD test consists of two steps. In step I, 4 μL of antibiotic solution (100 μg/mL RIF) was used to impregnate a sterilized filter paper disk of 6 mm diameter, which was then placed on LB agar plates with ~$5 \times 10^7$ cfu of bacteria. After allowing diffusion for 15 min, the plates were incubated at 37 °C. MIC comparison was done by measuring the radius of the inhibition zone. In step II, after incubation at 37 °C for 12–16 h, a nutrition disk with 5 μL of 40%-glucose was placed, and the plates were incubated again at 37 °C.

## Label-free proteomics

Label-free quantitative proteomics was conducted to investigate proteomic changes among the Non-stressed, RelE$_{low}$, and RelE$_{high}$ groups. (1) nano LC-MS/MS analysis. For each sample, 200 ng of total peptides were separated and analyzed with a nano-UPLC (nanoElute2) coupled to a timsTOF Pro2 instrument (Bruker) with a nano-electrospray ion source. Separation was performed using a reversed-phase column (PePSep C18, 1.9 μ, 75 μ × 15 cm, Bruker, Germany). Mobile phases were H2O with 0.1% FA (phase A) and ACN with 0.1% FA (phase B). Separation of the sample was executed with a 60 min gradient at 300 nL/min flow rate. Gradient B: 2% for 0 min, 2–22% for 45 min, 22–37% for 5 min, 37–80% for 5 min, 80% for 5 min. The mass spectrometer adopts DDA PaSEF mode for DDA data acquisition, and the scanning range is from 100 to 1700 m/z for MS1. During PASEF MS/MS scanning, the impact energy increases linearly with ion mobility, from 20 eV (1/K0 = 0.6 Vs/cm²) to 59 eV (1/K0 = 1.6 Vs/cm²). (2) SpectroMine database search Vendor's raw MS files were processed using SpectroMine software (4.1.230421.52329) and the built-in Pulsar

search engine. MS spectra lists were searched against their species-level UniProt FASTA databases (uniprotkb_Escherichia_coli_reviewed_2023_07. fasta), Carbamidomethyl [C] as a fixed modification, oxidation (M), and acetyl (Protein N-term) as variable modifications. Trypsin was used as proteases. A maximum of two missed cleavage(s) was allowed. The false discovery rate (FDR) was set to 0.01 for both PSM and peptide levels. Peptide identification was performed with an initial precursor mass deviation of up to 20 ppm and a fragment mass deviation of 20 ppm. All the other parameters were reserved as default.

## Data availability

All data needed to evaluate the conclusions in the paper are present in the paper and/or the Supplementary Materials.

The source data of this paper are collected in the following database record: biostudies:S-SCDT-10_1038-S44320-025-00174-6.

## Peer review information

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

## Acknowledgements

This work was supported by grants from the National Natural Science Foundation of China (Grant Numbers 21877091, 21934004, and 21627811), the National Key R&D Program of China (2021YFA0909400), and the Natural Science Foundation of Fujian Province, China (2022J01024).

## Author contributions

**Lina Wu**: Conceptualization; Resources; Formal analysis; Supervision; Funding acquisition; Methodology; Writing—original draft; Project administration; Writing—review and editing. **Qingqing Wang**: Data curation; Validation; Investigation; Visualization; Methodology; Writing—original draft. **Xinyi Hong**: Data curation; Software; Formal analysis; Visualization. **Xueer Cai**: Data curation; Formal analysis; Validation; Investigation; Visualization. **Litinghui Zhang**: Data curation; Validation; Investigation; Visualization. **Min Li**: Investigation. **Mingkai Wu**: Investigation. **Thomas K Wood**: Supervision; Writing—review and editing. **Xiaomei Yan**: Supervision; Project administration; Writing—review and editing.

Source data underlying figure panels in this paper may have individual authorship assigned. Where available, figure panel/source data authorship is listed in the following database record: biostudies:S-SCDT-10_1038-S44320-025-00174-6.

## Disclosure and competing interests statement

The authors declare the following competing financial interest(s): XY declares competing financial interest as a cofounder of NanoFCM Inc., a company committed to commercializing the nano-flow cytometry (nFCM) technology.

