## [Peer Review File · Molecular Systems Biology]

Single-cell analysis reveals critical toxin/antitoxin ratio triggering persister resuscitation

Lina Wu, Qingqing Wang, Xinyi Hong, Xurer Cai, Litinghui Zhang, Min Li, Mingkai Wu, Thomas Wood, and Xiaomei Yan

Corresponding author(s): Lina Wu (alina1222@xmu.edu.cn), Xiaomei Yan (xmyan@xmu.edu.cn)

Review Timeline:

Submission Date:	6th May 25
Editorial Decision:	9th Jun 25
Revision Received:	6th Sep 25
Editorial Decision:	10th Oct 25
Revision Received:	3rd Nov 25
Editorial Decision:	17th Nov 25
Revision Received:	17th Nov 25
Accepted:	20th Nov 25

Editor: Yehu Moran

Transaction Report:

9th Jun 2025

Manuscript Number: MSB-2025-13096-T

Title: Single-cell analysis reveals critical toxin/antitoxin ratio triggering persister resuscitation

Author: Lina Wu

Qingqing Wang

Xinyi Hong

Min Li

Litinghui Zhang

Xurer Cai

Mingkai Wu

Thomas Wood

Xiaomei Yan

Dear Dr. Wu,

Thank you again for submitting your work to Molecular Systems Biology. We have now heard back from the three reviewers who agreed to evaluate your manuscript. As you will see from the reports below, the referees find the topic of your study of potential interest. They raise, however, various concerns regarding your work, which preclude its publication in its present form. I am especially concerned about the comments made by Reviewer #1 regarding the potential limitations of your experimental system and I would like to see your response to these specific claims. Yet, in light of the considerable enthusiasm raised by the two other reviewers I have decided to provide you with the opportunity to revise your manuscript and address the concerns.

When you resubmit your manuscript, please download our CHECKLIST (<https://bit.ly/EMBOPressAuthorChecklist>) and include the completed form in your submission.

Please note that the Author Checklist will be published alongside the paper as part of the transparent process (<https://www.embopress.org/page/journal/17444292/authorguide#transparentprocess>).

If you feel you can satisfactorily deal with all the points listed by the reviewers, you are welcome to submit a revised version of your manuscript. Please attach a covering letter giving details of the way in which you have handled each of the points raised by the reviewers. A revised manuscript will be once again subject to review and you probably understand that we can give you no guarantee at this stage that the eventual outcome will be favorable. Should you have any specific questions regarding our process or if you would like to consult regarding any aspects of preparing your revision, please do not hesitate contacting me at my email: y.moran@molsystbiol.org

Yours sincerely,

Yehu Moran

Academic Editor

Molecular Systems Biology

We realize that it is difficult to revise to a specific deadline. In the interest of protecting the conceptual advance provided by the work, we recommend a revision within 3 months (7th Sep 2025). Please discuss the revision progress ahead of this time with the editor if you require more time to complete the revisions. Use the link below to submit your revision:

IMPORTANT: When you send your revision, we will require the following items:

1. the manuscript text in LaTeX, RTF or MS Word format
2. a letter with a detailed description of the changes made in response to the referees. Please specify clearly the exact places in the text (pages and paragraphs) where each change has been made in response to each specific comment given
3. three to four 'bullet points' highlighting the main findings of your study
4. a short 'blurb' text summarizing in two sentences the study (max. 250 characters)
5. a 'thumbnail image' (550px width and max 400px height, Illustrator, PowerPoint or jpeg format), which can be used as 'visual title' for the synopsis section of your paper.
6. Please include an author contributions statement after the Acknowledgements section (see <https://www.embopress.org/page/journal/17444292/authorguide>)
7. Please complete the CHECKLIST available at (<https://bit.ly/EMBOPressAuthorChecklist>).

Please note that the Author Checklist will be published alongside the paper as part of the transparent process (<https://www.embopress.org/page/journal/17444292/authorguide#transparentprocess>).

See also figure legend guidelines: <https://www.embopress.org/page/journal/17444292/authorguide#figureformat>

9. Please note that corresponding authors are required to supply an ORCID ID for their name upon submission of a revised manuscript (EMBO Press signed a joint statement to encourage ORCID adoption).

(<https://www.embopress.org/page/journal/17444292/authorguide#editorialprocess>)

Currently, our records indicate that the ORCID for your account is 0000-0002-7106-4752.

Link Not Available

11. Include a Reagents and Tools Table as part of the Methods section, which can be downloaded from our author guidelines (<https://www.embopress.org/page/journal/17444292/authorguide#structuredmethods>)

*** PLEASE NOTE *** As part of the EMBO Press transparent editorial process initiative (see our Editorial at <https://dx.doi.org/10.1038/msb.2010.72>), Molecular Systems Biology publishes online a Review Process File with each accepted manuscripts. This file will be published in conjunction with your paper and will include the anonymous referee reports, your point-by-point response and all pertinent correspondence relating to the manuscript. If you do NOT want this File to be published, please inform the editorial office at contact@molsystbiol.org within 14 days upon receipt of the present letter.

Reviewer #1:

The manuscript investigates bacterial persistence by analyzing the RelBE toxin-antitoxin system using single-cell nano-flow cytometry and fluorescence labeling. The authors show that a T/A ratio of 1.0 marks a key threshold: higher ratios are associated with dormancy (persister formation), while lower ratios correspond to bacterial regrowth. They show that elevated RelE levels correlate with deeper dormancy, and that toxin degradation is necessary for resuscitation. The study introduces a method to quantify protein expression and monitor cell division, offering some insights into how antibiotic tolerant cells form and recover. Despite some technical strengths, the study has several issues that limit its current suitability for publication. It focuses only on the RelBE system, without testing whether the findings apply to other TA systems, making the conclusions narrow. The use of artificial tags to measure protein expression could alter native protein behavior, and no genetic validations are used to prove some conclusive statements. Additionally, the study examines only a limited set of antibiotics, all of which are primarily bacteriostatic agents that target transcription or translation machinery. As a result, it is difficult to distinguish whether the observed dormancy is a specific persister phenotype or simply a general growth-arrested state induced by bacteriostatic stress. Since such antibiotics suppress growth without killing, the apparent increase in dormant cells within the bulk population may not reflect true persister formation. This makes the study's scope narrow and raises questions about the generalizability of the conclusions across other antibiotic classes and stress conditions. Some of my major comments are as follows:

1. While the authors state that TC tagging of RelE and RelB does not affect growth or persistence, this alone does not confirm that the tags have no impact on the expression levels, stability, or function of the proteins. Functional validation of the tagged proteins is critical, particularly in a tightly regulated system like RelBE. One way to assess this would be to overexpress both the tagged and untagged versions of RelE and RelB and compare their effects on cell dormancy; any functional disruption caused by tagging would likely result in divergent phenotypes. Additionally, comparing mRNA and protein levels of tagged versus untagged constructs using qRT-PCR and targeted proteomics, respectively, would help determine whether tagging affects expression or protein stability. Ideally, a time-course targeted proteomic analysis would provide direct evidence of the tagged proteins' degradation dynamics and confirm whether their behavior accurately reflects that of their native counterparts. Without these validations, it remains uncertain whether the observed single-cell expression patterns truly represent native TA system dynamics.

2. The authors claim that the expression of RelE and RelB can be detected in single cells with high sensitivity using the nFCM-T/A-TC-FIAsH strategy (see lines 101-102); however, this conclusion is incomplete. While the detection of fluorescence indicates that tagged proteins are being measured, it does not confirm that the fluorescence signal accurately reflects native protein

abundance or behavior. Fluorescence intensity alone cannot distinguish between properly folded, functional proteins and artifacts caused by tagging, altered folding, or differential degradation. Although the observation that RelB levels exceed RelE (which is supported by existing research), additional controls are needed to validate that the fluorescent signal corresponds to biologically meaningful expression. These should include comparisons of tagged versus untagged proteins using qRT-PCR for mRNA levels and targeted proteomics for absolute protein quantification (see my comment above).

3. Although the manuscript sets out to investigate the molecular mechanisms underlying rare persister cell formation, its experimental design does not align with this stated goal. In the introduction, the authors emphasize that persisters constitute a rare subpopulation that has traditionally been obscured in bulk population studies, and they explicitly critique previous work for failing to resolve the behavior of these cells at single-cell resolution. However, the experimental validation of their single-cell quantification method relies almost entirely on exposure to bacteriostatic antibiotics (e.g., rifampicin) which are known to inhibit transcription or translation and induce a global dormancy-like state across the entire population, not just in persisters. As a result, the elevated T/A ratios and expression patterns they report are likely reflective of generalized stress responses, not the behavior of true persister cells. This disconnect raises a fundamental question about the novelty and relevance of the study. If the authors are effectively measuring bulk responses to bacteriostatic stress, then their single-cell assay is not revealing insights that couldn't already be captured using conventional bulk techniques, such as western blotting or RT-qPCR. More importantly, the study does not apply the method to bactericidal antibiotics (without pretreatments), which are required to selectively eliminate growing cells and thereby isolate and study bona fide persisters. Although the authors suggest their method can detect rare cells, they do not demonstrate its utility in conditions where persisters are the only survivors. This undermines the key claim of the manuscript that their approach provides a novel and sensitive window into persister biology. Established methods such as fluorescent reporters, FACS, and microscopy have already been applied successfully to study these rare cells. Unless the authors directly apply their technique to distinguish and track true persisters under bactericidal stress, it remains unclear what unique advantage their method offers.

4. While the authors claim to investigate the molecular basis of persister formation, the experimental design undermines that goal. Specifically, their key persistence assays involve pre-treating *E. coli* cultures with rifampicin, a bacteriostatic transcription inhibitor, prior to challenging the cells with the bactericidal antibiotic ampicillin. This pretreatment artificially induces a dormant-like state in the bulk population, which is well known to increase antibiotic tolerance. As such, the resulting ampicillin survival may reflect a general, stress-induced growth arrest rather than the behavior of spontaneously arising, rare persister cells. This is a critical distinction, as the study is framed around resolving the biology of naturally occurring persisters, not induced tolerance. To make a meaningful contribution to persister biology, the authors should repeat key experiments using bactericidal antibiotics directly, without pretreatment. These should include diverse classes such as β -lactams, fluoroquinolones, and aminoglycosides, which effectively eliminate growing cells and allow true persisters to be isolated. Only under such conditions can the authors convincingly demonstrate that their single-cell quantification method captures the behavior of bona fide persister cells, rather than stress-arrested bulk populations. Without these additional experiments, the study does not deliver on its central premise.

5. Figure 2: At higher rifampicin concentrations, overall protein expression is expected to be inhibited due to transcriptional shutdown, which likely explains the observed reduction in protein levels. The significant decline in RelB abundance, in particular, may also result from its known instability and rapid degradation, as antitoxins like RelB are generally labile. Please verify whether the decrease in RelB levels under high rifampicin stress reflects transcriptional inhibition, increased degradation, or both. Given that chloramphenicol halts translation, I would expect no new RelB synthesis at high concentrations. Since RelB is a known labile antitoxin, it should undergo degradation in the absence of ongoing synthesis, particularly via Lon protease. Therefore, a reduction in fluorescence signal (reflecting RelB levels) would be anticipated under these conditions. However, an opposite effect is reported; this is interesting. Chloramphenicol might be affecting protease levels (I would suggest the authors to check these mechanisms). The authors also report a significant increase in fluorescence signal at higher tetracycline concentrations, which they interpret as elevated expression of RelE and/or RelB. However, this interpretation may be confounded by the fact that tetracycline itself is fluorescent, with an emission spectrum overlapping that of the FIAsh dye used in their detection system. Without appropriate controls, it is not possible to determine whether the observed fluorescence increase truly reflects protein abundance or is simply due to tetracycline autofluorescence.

6. Figure 3: Not all non-lysed cells following antibiotic treatment can be confidently classified as persisters; some may be dead or in a viable but non-culturable (VBNC) state. It is essential to verify whether the measured or analyzed subpopulations truly represent persister cells. One straightforward approach would be to examine sorted cells under time-lapse microscopy to confirm their ability to resume growth and form colonies after antibiotic removal. Given that the authors are already monitoring resuscitation kinetics, it is unclear why microscopy was not employed during this phase to directly validate cell viability and regrowth. Incorporating such imaging-based validation would significantly strengthen the claim that the tracked cells are functionally persistent and not merely inert survivors.

7. Figure 6-7: Could the non-growing cells identified in Figures 6F, G, and H represent VBNC cells? Another important point is that the authors should consider quantifying the absolute number of non-growing cells and calculating their ratio relative to the initial total population, rather than relying solely on percentages derived from flow cytometry plots. Percentages alone can be misleading (especially in dynamic cultures) because fast-growing subpopulations can rapidly dominate, artificially reducing the proportion of non-dividing cells even if their absolute number remains unchanged. Thus, a decline in the percentage of dormant cells may reflect population expansion rather than true resuscitation. To distinguish whether the impact of RIF is on lag phase

duration, cell proliferation rate, or the number of growing cells, the authors should directly quantify both proliferating and non-proliferating subpopulations. Some flow cytometers can measure both volume and event count, which can help estimate cell numbers; if not, counting beads should be used. This would allow the authors to determine whether RIF pre-treatment delays growth initiation, reduces the number of growth-competent cells, or alters proliferation kinetics. In addition, correlating these measurements with microscopy-based viability and outgrowth tracking would further validate their observations.

Reviewer #2:

In this elegant work, a novel tool was developed to analyze the activity of the toxin-antitoxin system at a single-cell level. The authors developed a nano-flow cytometer (nFCM) with superior sensitivity for simultaneous fluorescence and light scattering measurements. This method allowed the authors to measure, in single-cell resolution, the response of TA to antibiotics with accuracy and precision and to determine the physiological mechanism by which TAs could mediate persistence. The results, as demonstrated in Figure 3-7, show a linear correlation between the T/A ratio and the number of persister cells as well as demonstrate that resurrection may reflect decrease in the levels of RelE. These results are important and novel, as they allow us to accurately predict cellular events leading to the formation of persister cells, which do not respond to antibiotic, and the increase in their frequency following antibiotic treatment.

I have a few questions and comments which could be better addressed in this work.

Major comments:

1. The literature indicates that the instability of the antitoxin and the formation of toxin-antitoxin complex play a crucial role in the functionality of the toxin-antitoxin (TA) system. The presence of the tag does not affect growth. Did the authors rule out the potential impact of antitoxin degradation or toxin-antitoxin interactions that could influence the interpretation of the results? Additionally, there is no strong phenotype observed in LB for the deletion of the relBE system, so it is plausible that the experiment presented in Figure S1 is insufficient to evaluate the effect of the flash-TC tags.
2. At low RIF pressure, the authors observed increased expression of both RelE and RelB, with their levels peaking at the minimum inhibitory concentration (MIC). Can the authors comment on whether this is due to transcriptional or post-transcriptional regulation? Both the toxin and antitoxin are induced and expressed from the same promoter; thus, the effect appears to be transcriptional regulation, which does not contradict differential degradation. This is consistent with the model in Figure 2. As rifampicin inhibits transcription (e.g.), this is counterintuitive. This is not the case for translation inhibitors
3. "When T/A {greater than or equal to} 1.0, RelE expression increases while RelB expression decreases". I have looked and I do not see an indication to an additional promoter before relE.
4. A control for self-fluorescence should be included for each antibiotic treatment to rule out alterations in basal level of fluorescence.
5. Can the authors incorporate the results in concentration >MIC with a live-dead stain? I am not sure that the T/A ratio should be calculated from dead cells.
6. RelE degrades mRNA with codon specificity ([https://www.cell.com/fulltext/S0092-8674\(02\)01248-5](https://www.cell.com/fulltext/S0092-8674(02)01248-5)). Can the authors include the analysis of putative RelE target codons within the transcripts encoding the differentially expressed proteins and their abundance between the with low and high RelE expression populations? Also, did the authors confirm the expected reduction of expressed proteins with high RelE expression?
7. Did the results from proteomic analysis confirm the flow cytometry results regarding the T/A ratio and the sorting of the population? The essays in the manuscript rely only on the tags, but this simple test can verify the results

Minor comments:

*Line 50: This is the first time that type II toxin-antitoxins are mentioned, please define them

*Line 66: In principle, flow cytometry can overcome both the small size of bacteria and the low expression levels. The development of image stream flow cytometry can also be helpful. Multiple examples were published in *Bacillus subtilis* and *Listeria* which are indeed bigger but still representative of flow cytometry as analytical tool. Please revise this statement

*Figure S1: I am a bit puzzled on the demonstration of the lack of toxicity in *E. coli* BW25113 (blue), while the text (L88) states that these tags were "constructed (in) two *E. coli* MG1655 strains.". I think that BW25113 is K-12 but not MG1665 as it carries additional mutations (F- DE(araD-araB)567 lacZ4787(del)::rrnB-3 LAM- rph-1 DE(rhaD-rhaB)568 hsdR514)

*Fig 1 legend: multiple fonts

*L245: RelE expression may be crucial not only for the induction of persister cells but also for their resuscitation. RelE expression or RelE/RelB ratio?

*Can the authors add a statistical test (Anova) to Fig 7D-E (it seems that the time is of significance)?

Reviewer #3:

This work investigates bacterial persisters, antibiotic-tolerant subpopulations that contribute to infection relapses and antibiotic resistance. The authors developed a novel nano-flow cytometry-based method to quantitatively analyze toxin (T) and antitoxin (A) levels of relBE TA system and monitor bacterial states at the single-cell level. A key finding is that in response to antibiotic stress bacteria T:A ratio changes through two distinct TA expression modes, with T:A = 1.0 identified as a critical threshold. A strong correlation was observed between whole-population bacterial survival and the average T:A ratio, suggesting that higher

T:A ratios could promote persistence. At the single-cell level, persisters show heterogeneous RelE expression, which correlates with the duration of dormancy. Authors propose that persister resuscitation involves a triphasic detoxification process, where the progressive degradation of RelE toxin reduces the T:A ratio, which enables cell proliferation when the ratio drops to a permissive threshold. Additionally, the authors developed a cell wall growth-based platform using wheat germ agglutinin to label bacteria, a technique that does not require genetic modification. Proteomic analysis of cells with various RelE levels revealed that persisters with high RelE levels have increased expression of transmembrane transporters related to stress response and drug efflux. The figures are clear and easy to follow, and the text is well-written and simple to understand.

The findings underscore the importance of using maximum antibiotic concentrations in treatment and provide valuable molecular insights into the dynamics of persistence transitions. Overall, I think that this study is compelling and contributes to the study of TA dynamics and its connection to antibiotic recalcitrance. It will likely attract interest of microbiologists studying bacterial survival mechanisms, such as resistance, tolerance, and persistence. My major and minor points are listed below.

Major points:

1. Authors state that TC tag addition did not affect bacterial growth or persistence (lines 92-94). While the TC tag is small and apparently has a minimal interference to protein structure and function, they did not test whether its addition to RelE toxin or RelB antitoxin, both being small proteins, alters their specific functions, such as mRNA cleavage activity of RelE, operon repression by RelB, or neutralization of RelE by RelB. The effect of TC on stability of RelE and RelB should also be analyzed. This is critical for the validity of the results.
2. A non-TA control protein labeled with a TC tag and exhibiting consistent TC-FIAsH fluorescence should be included. This control would confirm that differences in RelB and RelE protein levels are specific to the relBE TA system's dynamics rather than artifacts of TC tag labeling or fluorescence variability.
3. When discussing Figure 1C results (lines 103-106), the authors should validate nFCM-T/A-TC-FIAsH measurements of RelE and RelB levels using qRT-PCR (for mRNA) and Western blotting (for protein) to rule out biases in the platform, such as differential TC-FIAsH dye binding due to RelE or RelB protein conformation or tag accessibility.
4. The authors observe variable RelB and RelE fold increase at T:A ratios of 0.5-1.0, attributing this to variable synthesis and degradation rates for each protein. A recommended control to validate this model is a Δ lon protease mutant, which should prevent preferential RelB antitoxin degradation (PMID: 11717402), stabilizing T:A ratios and confirming its role in relBE dynamics.
5. In lines 219-221, it is stated that lower RelE levels challenge the view that high toxin expression defines persisters. However, *E. coli* contains multiple TA systems (e.g., mazEF, hipBA), and persisters with lower RelE could have elevated levels of other toxins (e.g., MazF, HipA). This possibility should be discussed.
6. In lines 389-391, the authors state that the average RelE:RelB ratio exceeded 1 at 120 min (Figure 7C and F), yet bacteria already started replicating. Figure S3B shows that levels of RelE continued to rise at 180 and 240 min, but the levels of RelB and RelE:RelB ratio for these time points are not shown in Figure 7D-F. This should be included. If the average RelE:RelB ratio was above 1 for these time points, then the question arises - how come bacteria were replicating?
7. A major limitation of the nFCM-T/A-TC-FIAsH technique is that, while RelE and RelB levels are measured at the single-cell level, their T:A ratio is calculated using median fluorophore intensities from two independent populations. This should be discussed, and the authors could propose future development of dual-labeling techniques (e.g., TC-FIAsH/ReAsH) to measure single-cell T:A ratios directly.

Minor points:

1. Authors should correct their language throughout the text as proteins do not get expressed, genes are.
2. Bacterial strain construction section (lines 458-475) lacks a citation for the method used.
3. There is an inconsistency in strain nomenclature (In line 90 there is "MG1655", whereas in lines 459, 466, and 477 a "BW25113" name is used).
4. Proteomic profiling method description is missing.
5. Line 142 - Can authors clarify the " $[(\text{RelB}) > (\text{RelE}) > (\text{RelB}_2)]$ "? This is confusing.
6. Are Figure 4F average resuscitation time values? If so, SD or SEM values are missing. Also, there is no statistical analysis performed.

7. Similarly, error bars are missing in Figures 6D, G and H, and 7B.
8. There is a typo in Fig. S1C time axis
9. Acronym "CGSC" in methods section (line 460) requires clarification
10. There are multiple typos throughout the text that require addressing, also various fonts appear in the text (i.e. line 115)

Molecular Systems BiologySeptember 6th, 2025

Dear Editor:

Re: Manuscript ID. MSB-2025-13096-T

Please find attached a revised version of our manuscript "Single-cell analysis reveals critical toxin/antitoxin ratio triggering persister resuscitation". We sincerely thank you for the opportunity to revise our work and for taking the time to guide the review process. We also thank the three reviewers for their careful reading, positive assessment, and constructive suggestions. These comments have helped us to significantly improve the quality of the manuscript.

In response to the reviewer's suggestions, we performed additional experiments to further support our conclusions. Using Data-Independent Acquisition (DIA) proteomics, we demonstrated that genomic integration of the TC tags does not perturb global proteome homeostasis. This conclusion is bolstered by correlation coefficients greater than 0.97 for all sample pairs in the correlation heatmap, which has been included in the Supporting Information. Moreover, to confirm that the targeted cells were persisters rather than in a viable but non-culturable (VBNC) state, we monitored the resuscitation of the four types of persisters on solid agar plates. Over time, the colony counts of all four persister groups recovered to levels comparable to those of untreated control cells, confirming their identity as persisters. Additionally, our flow cytometry-based method effectively differentiated their persistence levels within 1 to 3 hours- significantly faster than traditional plate-based growth monitoring, which typically requires more than 10 hours.

Because additional experiments were conducted in the revised manuscript, Litinghui Zhang and Xueer Cai were moved forward in the author list for their contribution. In the following, we provide a point-by-point answers to the comments of the reviewers and describe the major revisions that have been implemented in the manuscript. We have submitted two versions of the revised paper: one with all the changes highlighted and one with no changes visible.

We hope that our manuscript can now be accepted for publication in *Molecular Systems Biology*.

Yours sincerely,

On behalf of all the authors,

Dr. Lina Wu

Associate Professor
Department of Chemical Biology
College of Chemistry & Chemical Engineering
Xiamen University
Xiamen, Fujian 361005, China
FAX: +86-592-218-9959
E-mail: alina1222@xmu.edu.cn

Response to the comments of Reviewer 1

The manuscript investigates bacterial persistence by analyzing the RelBE toxin-antitoxin system using single-cell nano-flow cytometry and fluorescence labeling. The authors show that a T/A ratio of 1.0 marks a key threshold: higher ratios are associated with dormancy (persister formation), while lower ratios correspond to bacterial regrowth. They show that elevated RelE levels correlate with deeper dormancy, and that toxin degradation is necessary for resuscitation. The study introduces a method to quantify protein expression and monitor cell division, offering some insights into how antibiotic tolerant cells form and recover. Despite some technical strengths, the study has several issues that limit its current suitability for publication. It focuses only on the RelBE system, without testing whether the findings apply to other TA systems, making the conclusions narrow. The use of artificial tags to measure protein expression could alter native protein behavior, and no genetic validations are used to prove some conclusive statements. Additionally, the study examines only a limited set of antibiotics, all of which are primarily bacteriostatic agents that target transcription or translation machinery. As a result, it is difficult to distinguish whether the observed dormancy is a specific persister phenotype or simply a general growth-arrested state induced by bacteriostatic stress. Since such antibiotics suppress growth without killing, the apparent increase in dormant cells within the bulk population may not reflect true persister formation. This makes the study's scope narrow and raises questions about the generalizability of the conclusions across other antibiotic classes and stress conditions. Some of my major comments are as follows:

Response: We greatly appreciate your thoughtful feedback and insightful suggestions, which will undoubtedly help us clarify and enhance the presentation of our findings in the manuscript.

a. It focuses only on the RelBE system, without testing whether the findings apply to other TA systems, making the conclusions narrow.

– We appreciate the reviewer's insightful comment regarding the scope of our study. We focused specifically on the RelBE system due to its established role in bacterial persistence, and its experimental advantages in single-cell quantification. We acknowledge that extending the findings to other TA systems would strengthen the broader implications, which are needed for future work.

b. The use of artificial tags to measure protein expression could alter native protein behavior, and no genetic validations are used to prove some conclusive statements.

– We appreciate the reviewer's emphasis on the importance of rigorous functional validation of TC-tagged proteins. In response to this concern, we present four integrated lines of evidence demonstrating that TC-tagged RelBE retains its native functionality. For detailed information, please refer to our response to question 1 below for this reviewer 1) Preservation of native state via genomic integration, page 6; 2) Minimalist tag design with prior validation refer to some specific paper, etc.; 3) Structural validation of functional TC-tag insertion in RelB/RelE complex, Figure 3; 4) Systems-level validation through DIA proteomics, Figure 4.

c. Additionally, the study examines only a limited set of antibiotics, all of which are primarily bacteriostatic agents that target transcription or translation machinery.

– We acknowledge the importance of studying naturally-occurring persister cells and their mechanisms. However, antibiotic-induced persisters that form under therapeutic stress also pose a crucial clinical challenge. Direct treatment with bactericidal antibiotics predominantly targets naturally-occurring persisters; however, their limited abundance makes systematic study technically challenging. To facilitate robust methodological development, we therefore adopted a model using bacteriostatic antibiotic pretreatment to generate a higher yield of treatment-induced persister cells. This population remains highly clinically relevant and enables more reproducible analysis under controlled conditions. We emphasize that both types of persister cells are medically significant. Our approach provides a tractable system to establish high-throughput screening methods that can later be extended to naturally occurring persisters through enrichment strategies. This foundational work will support future studies aimed at uncovering mechanisms and therapeutic strategies against both natural and induced persister cells.

To clarify this focus, we have added the term “bacteriostatic” before “antibiotic” in line 20 of the abstract and “two additional bacteriostatic antibiotics:” in line 154 of the results, thereby more accurately reflecting the core contribution of our work.

d. As a result, it is difficult to distinguish whether the observed dormancy is a specific persister phenotype or simply a general growth-arrested state induced by bacteriostatic stress.

– We appreciate the reviewer's insightful question. We have demonstrated the characteristics of persister cells through multiple approaches. The quantitative persistence assay results, shown in Fig. 3a of the manuscript, together with findings from the disk diffusion assay and the Tolerance Disk Test (TDTTest) in Fig. 3b, as well as the doubling time (Figure 1a below), killing curves (Figure 1b below) and growth monitoring on agar plates (Figure 2 below), provide strong evidence of persistence. Persisters that emerged under different antibiotic conditions were shown to have doubling times similar to untreated bacteria, as seen in Figure 1a, which agrees with previous single cell results for persister resuscitation from multiple labs and is a primary phenotype of persister cells (e.g. doi:10.1111/1462-2920.14093).

As illustrated by the killing curve in Figure 1b, persisters subjected to various RIF concentrations exhibit increased tolerance to AMP relative to normal cells. Notably, persisters obtained at low RIF concentrations (0.5 MIC and 1 MIC) demonstrate higher tolerance than those from 5 MIC and 10 MIC. This finding aligns with the single-cell-level resuscitation experiments in Fig. 6F in manuscript, where persisters from lower concentrations exhibited deeper dormancy.

Furthermore, we quantified the persisters extracted under these conditions using nFCM, inoculated equivalent numbers onto LB plates alongside normal cells, and tracked their growth over time. As depicted in Figure 2, persisters from 5 MIC and 10 MIC conditions formed visible colonies by 9 hours, whereas those from 0.5 MIC and 1 MIC conditions did so by 13 and 11 hours, respectively. Ultimately, the number of colonies generated under all four conditions equaled that of untreated control cells, as listed in Table 1, thereby further confirming their identity as persisters rather than in a viable but non-culturable (VBNC) state.

Figure 1. Bacterial persistence measurement for persisters purified from cells pretreated with 0.5, 1, 5 and 10 MIC RIF followed by AMP treatment, and total cells cultivated under non-stressed condition as a negative control. (a) Bar graph showing the average doubling times when bacteria grow in the exponential phase. (b) Frequency of persister formation in bacteria over time, determined by antibiotic (ampicillin) susceptibility measurements. Bars indicate the mean of at least three independent experiments; error bars represent the standard deviation (SD).

Figure 2. Plate culture monitoring of persister resuscitation for the four persister conditions during resuscitation over time.

Table 1 The Number of Colonies Produced by Four Conditions of Persisters

Group	1	2	3	4	Mean	Concentration/ (CFU/mL)
0 MIC	124	134	130	133	130.25 ± 4.50	1.30E+07
0.5 MIC	115	119	121	120	118.75 ± 2.63	1.19E+07
1.0 MIC	125	124	112	124	121.25 ± 6.18	1.21E+07
5.0 MIC	129	127	128	130	128.50 ± 1.29	1.29E+07
10.0 MIC	120	118	129	118	121.25 ± 5.25	1.21E+07

1. While the authors state that TC tagging of RelE and RelB does not affect growth or persistence, this alone does not confirm that the tags have no impact on the expression levels, stability, or function of the proteins. Functional validation of the tagged proteins is critical, particularly in a tightly regulated system like RelBE. One way to assess this would be to overexpress both the tagged and untagged versions of RelE and RelB and compare their effects on cell dormancy; any functional disruption caused by tagging would likely result in divergent phenotypes. Additionally,

comparing mRNA and protein levels of tagged versus untagged constructs using qRT-PCR and targeted proteomics, respectively, would help determine whether tagging affects expression or protein stability. Ideally, a time-course targeted proteomic analysis would provide direct evidence of the tagged proteins' degradation dynamics and confirm whether their behavior accurately reflects that of their native counterparts. Without these validations, it remains uncertain whether the observed single-cell expression patterns truly represent native TA system dynamics.

– We appreciate the reviewer's emphasis on rigorous functional validation of TC-tagged proteins. We agree that protein tags may perturb biological systems, which is why our experimental design prioritized maximal physiological relevance. Below we present integrated evidence demonstrating that TC-tagged RelBE retains native functionality, including the suggested proteomic experiment:

i. Preservation of Native State via Genomic Integration

To maintain authentic cellular stoichiometry and regulation, TC tags were genomically integrated at the native *relBE* locus (bacterial strains construction details in Methods), avoiding plasmid-based overexpression artifacts.

ii. Minimalist Tag Design with Prior Validation

The 12-amino acid TC tag (1.5 kDa) represents <14% of RelE's molecular weight (11.2 kDa), and about 15% of RelB's molecular weight (9.8 kDa). It has already been successfully implemented in various essential proteins (e.g. 2010, Nat. Protoc., doi: 10.1038/nprot.2010.129; 2016, J. Am. Chem. Soc., doi: 10.1021/jacs.6b03422; 2017, FASEB J, doi: 10.1096/fj.201700058rrrr).

iii. Structural Validation of Functional TC-Tag Insertion in RelB/RelE Complex

The three-dimensional structural analysis via AlphaFold (as shown in Figure 3 below), confirms that inserting a TC-tag at the N-terminus of RelE and the C-terminus of RelB does not alter the protein's structure nor affect the various interaction modes of RelB and RelE depicted in Figure 3. The TC-tag is fully extended and exposed at the periphery of the three-dimensional structure, whether on the monomers of RelE and RelB or on their interaction complexes, ensuring it does not interfere with specific binding to biarsenical dyes.

Figure 3. Structural Validation of Functional TC-Tag Insertion in RelB/RelE Complex

iv. Systems-level Validation by DIA Proteomics

We further conducted Data-Independent Acquisition (DIA) proteomics on wild-type *E. coli* BW25113 (WT), genomic RelE-TC strain (*relE^{TC}*), and genomic RelB-TC strain (*relB^{TC}*), respectively. As illustrated in Figure 4 below, the correlation heatmap indicates no significant difference between wild-type and TC-tagged strains, with correlation coefficients exceeding 0.97 for all sample pairs.

Figure 4. Correlation heatmap of wild-type *E. coli* BW25113 (WT), genomic RelE-TC strain (*relE^{TC}*), and genomic RelB-TC strain(*relB^{TC}*) via Deep DIA-MS.

Conclusion Statement: Combining *in silico* structural modeling, systems-level proteomics, and physiological phenotyping, we demonstrate that genomically integrated TC tags: (i) Do not alter RelE/RelB structure, stability, or interaction thermodynamics; (ii) Preserve global proteome homeostasis; (iii) Faithfully report native TA system behavior in single cells.

2. The authors claim that the expression of RelE and RelB can be detected in single cells with high sensitivity using the nFCM-T/A-TC-FIAsH strategy (see lines 101-102); however, this conclusion is incomplete. While the detection of fluorescence indicates that tagged proteins are being measured, it does not confirm that the fluorescence signal accurately reflects native protein abundance or behavior. Fluorescence intensity alone cannot distinguish between properly folded, functional proteins and artifacts caused by tagging, altered folding, or differential degradation. Although the observation that RelB levels exceed RelE (which is supported by existing research), additional controls are needed to validate that the fluorescent signal corresponds to biologically meaningful expression. These should include comparisons of tagged versus untagged proteins using qRT-PCR for mRNA levels and targeted proteomics for absolute protein quantification (see my comment above).

— We thank the reviewer for emphasizing the importance of validating the correlation between fluorescence protein abundance. Previous research has demonstrated that antitoxin RelB is translated at a higher rate than toxin RelE under natural growth conditions (2014, Cell, doi: 10.1016/j.cell.2014.02.033).

Furthermore, in our earlier studies, we demonstrated that nFCM provides high resolution for the quantitative measurement of low-level antitoxin in single bacterial cells (2019, ACS Chem. Biol. doi: 10.1021/acscchembio.9b00721). Given that the expression of toxin-antitoxin (TA) systems under their native promoters is generally low, we utilized micromolar to millimolar concentrations of IPTG to induce the production of antitoxin MqsA in BW25113 $\Delta mqsRA \Delta Km$ transformed with plasmid pCA24N-plac-mqsR-mqsA-TC (referred to as pLRAtc, with antitoxin expressed under the pT5-lac promoter featuring the TC tag). Consequently, the expression of *mqsA* could be regulated by the strong pT5-lac promoter through IPTG induction. As illustrated in the accompanying Figure 5 below, when the median fluorescence burst area was plotted against the IPTG concentration, an exponential curve was observed. At low micromolar concentrations of IPTG, the trend was steep, indicating that small increases in IPTG concentration resulted in significant enhancements in antitoxin production. This increase in fluorescence intensity began to plateau at 1 mM IPTG. These findings clearly demonstrate that nFCM enables high-resolution quantitative measurements of low-level antitoxin in single bacterial cells.

Additionally, our label-free quantitative proteomics analyses of non-stressed normal samples support this finding: quantification revealed that RelB is 4.19-fold more abundant than RelE, closely aligning with the 5.56-fold difference measured using the nFCM-T/A-TC-FIAsH strategy.

Figure 5. Flow cytometric analysis of antitoxin MqsA using the nFCM-TC-FIAsH strategy (2019, ACS Chem. Biol. doi: 10.1021/acscchembio.9b00721). (e) Histograms of fluorescence burst area distribution for BW25113 Δ mq sRA Δ Km/pLRAtc induced with 0, 0.05, 0.1, 0.3, 0.5, 0.8, 1, 2, 3, and 5 mM IPTG. The histograms were normalized to facilitate an easy comparison. (f) Dose response curve of MqsA production in single bacterial cells at various IPTG concentrations.

3. Although the manuscript sets out to investigate the molecular mechanisms underlying rare persister cell formation, its experimental design does not align with this stated goal. In the introduction, the authors emphasize that persisters constitute a rare subpopulation that has traditionally been obscured in bulk population studies, and they explicitly critique previous work for failing to resolve the behavior of these cells at single-cell resolution. However, the experimental validation of their single-cell quantification method relies almost entirely on exposure to bacteriostatic antibiotics (e.g., rifampicin) which are known to inhibit transcription or translation and induce a global dormancy-like state across the entire population, not just in persisters. As a result, the elevated T/A ratios and expression patterns they report are likely reflective of generalized stress responses, not the behavior of true persister cells. This disconnect raises a fundamental question about the novelty and relevance of the study. If the authors are effectively measuring bulk responses to bacteriostatic stress, then their single-cell assay is not revealing insights that couldn't already be captured using conventional bulk techniques, such as western blotting or RT-qPCR. More importantly, the study does not apply the method to bactericidal antibiotics (without pretreatments), which are required to selectively eliminate growing cells and thereby isolate and study bona fide persisters. Although the authors suggest their method

can detect rare cells, they do not demonstrate its utility in conditions where persisters are the only survivors. This undermines the key claim of the manuscript that their approach provides a novel and sensitive window into persister biology. Established methods such as fluorescent reporters, FACS, and microscopy have already been applied successfully to study these rare cells. Unless the authors directly apply their technique to distinguish and track true persisters under bactericidal stress, it remains unclear what unique advantage their method offers.

– In response to the question of whether the studied objects are indeed persisters, we provide evidence through various methods, including the persistence rate shown in Fig. 3a in manuscript, the disk diffusion assay and Tolerance Disk Test (TDtest) results in Fig. 3b in manuscript, the killing curve (Figure 1b up), and plate growth monitoring (Figure 2 up). As illustrated in the killing curve in Figure 1b up, persisters extracted at different RIF concentrations exhibit higher tolerance to AMP compared to normal cells. Furthermore, persisters obtained at lower RIF concentrations (0.5 MIC and 1 MIC) show greater tolerance than those extracted at 5 MIC and 10 MIC. This observation is consistent with the findings in Fig. 6F in manuscript, where resuscitation experiments at the single-cell level demonstrated deeper dormancy in persisters obtained at 0.5 MIC and 1 MIC compared to those obtained at higher concentrations.

Additionally, we quantified the persisters obtained under these conditions using nFCM, inoculated equal numbers of persisters and normal bacteria on plates, and monitored their growth over time. As shown in Figure 2 up, persisters from 5 MIC and 10 MIC conditions produced visible colonies by 9 hours, whereas those from 0.5 MIC and 1 MIC conditions exhibited colony growth only by 13 and 11 hours, respectively. Ultimately, the colony numbers grown under the four conditions were comparable to those of normal bacteria, further confirming these cells as persisters and validating our flow cytometry method for differentiating their persistence levels within 1 to 3 hours.

Concerning single-cell level analysis, we applied our nFCM strategy to examine all aspects of the study at the single-cell level, except for the experiment in Fig. 3 in manuscript, where we used bulk data to analyze the correlation between bacterial persistence and the T/A ratio under RIF stress. This is noted in the figure legend as “using whole population averages.” In Fig. 2 of the manuscript, where we discovered that bacteria elevate the T/A ratio through distinct expression modes to resist antibiotic stress, we used Median Fluorescence Intensity (MFI) obtained through the nFCM-T/A-TC-FIAsH strategy for analysis. MFI provides granular insights into cell populations with single-cell resolution, while bulk techniques report population-averaged behavior. They are complementary but address fundamentally different biological questions. In Fig. 4 of the manuscript, we examined the correlation between RelBE expression and persistence in single persister cells, revealing not only heterogeneous RelE expression but also highlighting that while most persisters exhibited high levels of RelE, some individual cells showed low levels comparable to sensitive cells. For Fig. 6F, we employed the nFCM-flWGA method to monitor the resuscitation of persisters pretreated with various RIF concentrations at the single-cell level. This analysis revealed deeper dormancy at lower antibiotic doses within 2 hours. Our single-cell analysis using nFCM-flWGA uncovers considerable heterogeneity in persister resuscitation, aligning with the

distinct, sequential stages of RelE reduction observed in single-cell analysis using the nFCM-T/A-TC-FIAsH strategy in Fig. 7.

In summary, the developed nFCM-T/A-TC-FIAsH and nFCM-flWGA methodologies provide a robust platform for high-throughput single-cell analysis of TA module expression and bacterial persister states in their native context. While microscopy methods can observe and analyze rare persister cells, as reported in previous studies, our nFCM method aims to achieve rapid, high-throughput analysis of persister cells for complementary insights. Due to the naturally low abundance of persisters, we adopted the RIF pretreatment method to obtain a substantial amount of persisters and purified them using the bactericidal antibiotic AMP to lyse sensitive bacteria. Our high-throughput single-cell analysis using nFCM-flWGA reveals significant heterogeneity in persister resuscitation, and our nFCM-T/A-TC-FIAsH analysis uncovers sequential stages of RelE reduction, which are challenging to detect with bulk or low-throughput single-cell analyses.

4. While the authors claim to investigate the molecular basis of persister formation, the experimental design undermines that goal. Specifically, their key persistence assays involve pre-treating *E. coli* cultures with rifampicin, a bacteriostatic transcription inhibitor, prior to challenging the cells with the bactericidal antibiotic ampicillin. This pretreatment artificially induces a dormant-like state in the bulk population, which is well known to increase antibiotic tolerance. As such, the resulting ampicillin survival may reflect a general, stress-induced growth arrest rather than the behavior of spontaneously arising, rare persister cells. This is a critical distinction, as the study is framed around resolving the biology of naturally occurring persisters, not induced tolerance. To make a meaningful contribution to persister biology, the authors should repeat key experiments using bactericidal antibiotics directly, without pretreatment. These should include diverse classes such as β -lactams, fluoroquinolones, and aminoglycosides, which effectively eliminate growing cells and allow true persisters to be isolated. Only under such conditions can the authors convincingly demonstrate that their single-cell quantification method captures the behavior of bona fide persister cells, rather than stress-arrested bulk populations. Without these additional experiments, the study does not deliver on its central premise.

– We fully acknowledge the fundamental importance of spontaneously generated persister cells as well as making the distinction between tolerance and persister cells. However, given the pervasive exposure of bacteria to antibiotics in clinical and environmental contexts, understanding antibiotic-induced tolerance mechanisms represents a clinically urgent priority in combatting treatment failures and antimicrobial resistance. Moreover, rifampicin-induced persister cells produced by us have been vetted 9 ways to confirm persistence (doi:10.1111/1462-2920.14093, by showing multi-

drug tolerance, rapid resuscitation of most cells within minutes with nutrients, dormancy based on no cell division when nutrients are absent, dormancy based on metabolic staining/cell sorting, no change in minimum inhibitory concentration, no resistance phenotype, similar morphology to ampicillin-induced persisters, similar resuscitation as ampicillin-induced persisters, and no spontaneous waking), far more than any other technique, including those without pretreatment. Hence, rifampicin-induced dormant cells have been demonstrated to be persisters, not tolerant cells. Furthermore, the biological insights made with pretreatment (e.g., resuscitation growth rates the same as wild-type cells, no spontaneous waking) have been confirmed, albeit years later, with cells that lack pretreatment. So we are studying bone fide persister cells. Finally, and practically, over 50 labs now utilize the pretreatment method to generate bone fide persister cells including the labs of Balaban in a *Nature* 2021 paper (Cm pretreatment, <https://doi.org/10.1038/s41586-021-04114-w>) along with Conlon (2023, eLife, CCCP pretreatment, <https://doi.org/10.7554/eLife.80246>), Heinemann (2022, Nat Commun, CCCP pretreatment, <https://doi.org/10.1038/s41467-022-28141-x>), Lewis (2012, paraquat pretreatment, doi:10.1128/AAC.00921-12), and Levin (2013, sub-MIC antibiotic pretreatment, doi: 10.1371/journal.pgen.1003123), so the method is well-established.

5. Figure 2: At higher rifampicin concentrations, overall protein expression is expected to be inhibited due to transcriptional shutdown, which likely explains the observed reduction in protein levels. The significant decline in RelB abundance, in particular, may also result from its known instability and rapid degradation, as antitoxins like RelB are generally labile. Please verify whether the decrease in RelB levels under high rifampicin stress reflects transcriptional inhibition, increased degradation, or both. Given that chloramphenicol halts translation, I would expect no new RelB synthesis at high concentrations. Since RelB is a known labile antitoxin, it should undergo degradation in the absence of ongoing synthesis, particularly via Lon protease. Therefore, a reduction in fluorescence signal (reflecting RelB levels) would be anticipated under these conditions. However, an opposite effect is reported; this is interesting. Chloramphenicol might be affecting protease levels (I would suggest the authors to check these mechanisms). The authors also report a significant increase in fluorescence signal at higher tetracycline concentrations, which they interpret as elevated expression of RelE and/or RelB. However, this interpretation may be confounded by the fact that tetracycline itself is fluorescent, with an emission spectrum overlapping that of the FIAsh dye used in their detection system. Without appropriate controls, it is not possible to determine whether the observed fluorescence increase truly reflects protein abundance or is simply due to tetracycline autofluorescence.

– Thank you for raising this important question. We observed that the expression of RelB varies under high concentrations of the three antibiotics studied. Specifically, under high concentrations of rifampicin (RIF), the expression of RelB decreases, whereas for the other two antibiotics, chloramphenicol (Cam) and tetracycline (TET), the expression of RelB increases. This differential expression aligns well with the T/A ratio and the associated mechanism of conditional cooperativity. When bacteria face >1 MIC of RIF, the T/A ratios are greater than one ($[RelE] > [RelB]$), indicating the presence of free RelE. As an endonuclease, RelE cleaves mRNA, thereby inhibiting translation. Consequently, the expression levels of both RelE and RelB are lower under high RIF pressure compared to low RIF pressure, with the expression of antitoxin RelB being notably reduced due to its instability in these conditions. This results in a decrease in RelB levels under high rifampicin stress due to both translation inhibition and degradation. For Cam, within the concentration range of 0.5 to 100 MIC, the expression of both RelE and RelB was consistently higher than in non-stressed conditions. The T/A ratio analysis (Fig. 2C-iii in manuscript) indicated that RelE/RelB fell within the transcriptional activation range of $0.5 < T/A < 1.0$, leading to increased expressions of both RelE and RelB. Regarding TET, at 50 and 100 MIC concentrations, both RelE and RelB expressions increased significantly, which is consistent with the T/A ratio falling within $0.5 < T/A < 1.0$. Thus, the different expression patterns of RelB under high concentrations of RIF, Cam, and TET perfectly correspond to the T/A ratio and the associated mechanism of conditional cooperativity.

Thank you for raising the important question regarding potential tetracycline autofluorescence interference. Both published literature (2023, Chem. Eng. J. <https://doi.org/10.1016/j.cej.2022.140492>) and our own fluorescence spectral analysis (Figure 6 below) confirm that tetracycline (TET) exhibits peak excitation at approximately 365 nm. Critically, our FIAsh-EDT₂ detection employed a 488 nm excitation wavelength. As demonstrated in Figure 5 below, TET shows negligible excitation at this wavelength. Therefore, under our experimental conditions: TET fluorescence is not excited; no measurable interference with FIAsh-EDT₂ detection occurs. We appreciate your vigilance on this technical consideration, which strengthens the validity of our fluorescence measurements.

Figure 6. The fluorescence excitation and emission spectra of tetracycline and FIAsh-EDT₂, and the transmission spectra of the bandpass filters used for the green (FL01- 520/35) and red (FF01-670/30) fluorescence detection.

6. Figure 3: Not all non-lysed cells following antibiotic treatment can be confidently classified as persisters; some may be dead or in a viable but non-culturable (VBNC) state. It is essential to verify whether the measured or analyzed subpopulations truly represent persister cells. One straightforward approach would be to examine sorted cells under time-lapse microscopy to confirm their ability to resume growth and form colonies after antibiotic removal. Given that the authors are already monitoring resuscitation kinetics, it is unclear why microscopy was not employed during this phase to directly validate cell viability and regrowth. Incorporating such imaging-based validation would significantly strengthen the claim that the tracked cells are functionally persistent and not merely inert survivors.

– As we explained in response to Question 3, we have used multiple methods to demonstrate the persistence properties of the bacterial persisters in our study. Specifically, for the persister cells used in the FACS sorting experiments (Manuscript Fig. 4) were purified under $1\times$ MIC RIF. As plate growth assays shown in Figure 2 (upper panel) and Table 1, these persister cells exhibit significantly delayed colony emergence compared to normal cells (11 hours vs. 6 hours, respectively). Despite this growth lag, the final colony counts under $1\times$ MIC rifampicin treatment ultimately match those of untreated normal cells. This combination of delayed growth kinetics with complete recovery provides definitive confirmation of their persister identity rather than compromised viability.

7. Figure 6-7: Could the non-growing cells identified in Figures 6F, G, and H represent VBNC cells? Another important point is that the authors should consider quantifying the absolute number of non-growing cells and calculating their ratio relative to the initial total population, rather than relying solely on percentages derived from flow cytometry plots. Percentages alone can be misleading (especially in dynamic cultures) because fast-growing subpopulations can rapidly dominate, artificially reducing the proportion of non-dividing cells even if their absolute number remains unchanged. Thus, a decline in the percentage of dormant cells may reflect population expansion rather than true resuscitation. To distinguish whether the impact of RIF is on lag phase duration, cell proliferation rate, or the number of growing cells, the authors should directly quantify both proliferating and non-proliferating subpopulations. Some flow cytometers can measure both volume and event count, which can help estimate cell numbers; if not, counting beads should be used. This would allow the authors to determine whether RIF pre-treatment delays growth initiation, reduces the number of growth-competent cells, or alters proliferation kinetics. In addition,

correlating these measurements with microscopy-based viability and outgrowth tracking would further validate their observations.

— Thank you for raising this important question. The plate growth assays shown in Figure 2 (upper panel) and Table 1 demonstrate that, despite an initial lag in growth, the final colony counts of persisters subjected to various rifampicin treatments ultimately match those of untreated normal cells. This evidence supports that the non-growing cells identified in Fig. 6F, 6G, and 6H in manuscript are persister cells rather than viable but non-culturable (VBNC) cells. Our high-sensitivity nFCM-flWGA strategy, which allows for the tracking of cell division by monitoring the decrease in fluorescence intensity per cell, enables real-time detection of resuscitating persisters at the single-cell level. Once a persister begins to divide, the fluorescence of the bacterium reduces by half. Consequently, we can observe the resuscitation of persisters in real-time by monitoring the decrease in fluorescence in the growing cell population, as shown in Fig. 6F region I of the manuscript. This method allows us to distinguish the impact of rifampicin on the duration of the lag phase. The high sensitivity of our nFCM-flWGA strategy further distinguishes the differences in dormancy depths among persisters exposed to 0.5, 1, 5, and 10 MIC of rifampicin within three hours, whereas conventional plate culture methods require much more time for differentiation. Persisters exposed to 5 MIC and 10 MIC conditions formed visible colonies within nine hours, while those from 0.5 MIC and 1 MIC conditions required 13 and 11 hours, respectively. Although microscopy-based viability and outgrowth tracking takes longer, it corroborates the observations made with our rapid nFCM-flWGA strategy.

Reviewer #2: In this elegant work, a novel tool was developed to analyze the activity of the toxin-antitoxin system at a single-cell level. The authors developed a nano-flow cytometer (nFCM) with superior sensitivity for simultaneous fluorescence and light scattering measurements. This method allowed the authors to measure, in single-cell resolution, the response of TA to antibiotics with accuracy and precision and to determine the physiological mechanism by which TAs could mediate persistence. The results, as demonstrated in Figure 3-7, show a linear correlation between the T/A ratio and the number of persister cells as well as demonstrate that resurrection may reflect decrease in the levels of RelE. These results are important and novel, as they allow us to accurately predict cellular events leading to the formation of persister cells, which do not respond to antibiotic, and the increase in their frequency following antibiotic treatment. I have a few questions and comments which could be better addressed in this work.

– We are grateful to the reviewer for the encouraging comments and constructive suggestions.

Major comments:

1. The literature indicates that the instability of the antitoxin and the formation of toxin-antitoxin complex play a crucial role in the functionality of the toxin-antitoxin (TA) system. The presence of the tag does not affect growth. Did the authors rule out the potential impact of antitoxin degradation or toxin-antitoxin interactions that could influence the interpretation of the results? Additionally, there is no strong phenotype observed in LB for the deletion of the *relBE* system, so it is plausible that the experiment presented in Figure S1 is insufficient to evaluate the effect of the flash-TC tags.

– We appreciate the reviewer's emphasis on rigorous functional validation of TC-tagged proteins. We agree that protein tags may perturb biological systems, which is why our experimental design prioritized maximal physiological relevance. Below we present integrated evidence demonstrating that TC-tagged RelBE retains native functionality:

a. Preservation of Native State via Genomic Integration

To maintain authentic cellular stoichiometry and regulation, TC tags were genomically integrated at the native *relBE* locus (bacterial strains construction details in Methods), avoiding plasmid-based overexpression artifacts.

b. Minimalist Tag Design with Prior Validation

The 12-amino acid TC tag (1.5 kDa) represents <14% of RelE's molecular weight (11.2 kDa), and about 15% of RelB's molecular weight (9.8 kDa). It has already been successfully implemented in various essential proteins. For example:

Hoffmann C, Gaietta G, Zurn A, Adams SR, Terrillon S, Ellisman MH, Tsien RY, Lohse MJ. Fluorescent labeling of tetracysteine-tagged proteins in intact cells. Nat Protoc, 2010, 5:1666-1677.

Walker AS, Rablen PR, Schepartz A. Rotamer-Restricted Fluorogenicity of the Bis-Arsenical ReAsH. J Am Chem Soc, 2016, 138: 7143-7150.

Bohl C, Pomorski A, Seemann S, Knospe AM, Zheng C, Krezel A, Rolfs A, Lukas J. Fluorescent probes for selective protein labeling in lysosomes: a case of alpha-galactosidase A. FASEB J, 2017, 31: 5258-5267.

c. Structural Validation of Functional TC-Tag Insertion in RelB/RelE Complex

The three-dimensional structural analysis (as shown in Figure 3 below) confirms that inserting a TC-tag at the N-terminus of RelE and the C-terminus of RelB does not alter the protein's structure nor affect the various interaction modes of RelB and RelE depicted in Figure 1. The TC-tag is fully

extended and exposed at the periphery of the three-dimensional structure, whether on the monomers of RelE and RelB or on their interaction complexes, ensuring it does not interfere with specific binding to biarsenical dyes.

Figure 1. Structural Validation of Functional TC-Tag Insertion in RelB/RelE Complex

d. Systems-level Validation by DIA Proteomics

We further conducted Data-Independent Acquisition (DIA) proteomics on Wild-type *E. coli* BW25113 (WT), Genomic RelE-TC strain (*relE^{TC}*), and Genomic RelB-TC strain (*relB^{TC}*), respectively. As illustrated in Figure 2 below, the correlation heatmap indicates no significant

difference between wild-type and TC-tagged strains, with correlation coefficients exceeding 0.97 for all sample pairs.

Figure 2. Correlation heatmap of wild-type *E. coli* BW25113 (WT), Genomic RelE-TC strain (*relE^{TC}*), and Genomic RelB-TC strain (*relB^{TC}*) via Deep DIA-MS.

Conclusion Statement: Combining *in silico* structural modeling, systems-level proteomics, and physiological phenotyping, we demonstrate that genomically integrated TC tags: (i) Do not alter RelE/RelB structure, stability, or interaction thermodynamics; (ii) Preserve global proteome homeostasis; (iii) Faithfully report native TA system behavior in single cells.

2. At low RIF pressure, the authors observed increased expression of both RelE and RelB, with their levels peaking at the minimum inhibitory concentration (MIC). Can the authors comment on whether this is due to transcriptional or post-transcriptional regulation? Both the toxin and antitoxin are induced and expressed from the same promoter; thus, the effect appears to be transcriptional regulation, which does not contradict differential degradation. This is consistent with the model in Figure 2. As rifampicin inhibits transcription (e.g.), this is counterintuitive. This is not the case for translation inhibitors.

—We agree with the reviewer that both the toxin and antitoxin are induced and expressed from the same promoter, thus, the effect appears to be transcriptional regulation, and this is consistent with the model in manuscript Fig. 2. When bacteria are challenged with ≤ 1 MIC RIF, T/A ratios between 0.5 to 1 ($[RelB] > [RelE] > [RelB_2]$) lead to the formation of $RelB_2RelE_2$, derepressing the promoter because $RelB_2RelE_2$ does not bind to the operator, which leads to the increased abundance of both RelE and RelB (Fig. 2A-i, ii).

3. "When T/A {greater than or equal to} 1.0, RelE expression increases while RelB expression decreases". I have looked and I do not see an indication to an additional promoter before relE.

—We confirm the absence of additional promoters upstream of *relE*, and this differential expression directly results from conditional cooperativity: Under $>1 \times$ MIC rifampicin, T/A ratios >1 release free RelE, which cleaves mRNAs and inhibits translation systemically, causing reduced expression of both *relE* and *relB*; crucially, RelB decreases more dramatically due to its intrinsic instability.

4. A control for self-fluorescence should be included for each antibiotic treatment to rule out alterations in basal level of fluorescence.

— We concur with the reviewer's suggestion that a control for self-fluorescence should be included for each antibiotic treatment to eliminate any potential changes in the baseline fluorescence levels. In the manuscript, we confirm that all median fluorescence intensities (MFIs) for *RelE^{TC}* and *RelB^{TC}* were determined by subtracting the background signal from the respective MFIs of *RelE^{TC}*

and $RelB^{TC}$ for each protein, specifically using the formulas $(MFI_{RelE^{TC}} - MFI_{wt})$ and $(MFI_{RelB^{TC}} - MFI_{wt})$, as shown in Fig. 1C in manuscript.

5. Can the authors incorporate the results in concentration $>MIC$ with a live-dead stain? I am not sure that the T/A ratio should be calculated from dead cells.

— We thank the referee for this insightful question. Heat-killed *E. coli* BW25113 (80°C, 20 min) served as our dead cell model. Using propidium iodide (PI)—a membrane-impermeant DNA intercalating dye that emits red fluorescence upon nucleic acid binding—we performed flow cytometry on: 1) heat-killed cells, and 2) live cells treated with 0, 1, 10, or 100× MIC rifampicin for 2 hours. As demonstrated in Figure 3, 99.8% of heat-killed cells were PI-positive. In rifampicin-treated populations, while PI-positivity increased modestly with concentration (0.8% at 0× MIC → 2.4% at 100× MIC), 97.6% of cells remained PI-negative at 100× MIC, confirming membrane integrity and demonstrating that the vast majority of bacteria survived this extreme antibiotic challenge.

Figure 3. Flow cytometric determination dead bacteria with propidium iodide (PI).

6. RelE degrades mRNA with codon specificity ([https://www.cell.com/fulltext/S0092-8674\(02\)01248-5](https://www.cell.com/fulltext/S0092-8674(02)01248-5)). Can the authors include the analysis of putative RelE target codons within the transcripts encoding the differentially expressed proteins and their abundance between the with low and high RelE expression populations? Also, did the authors confirm the expected reduction of expressed proteins with high RelE expression?

– We thank the reviewer for raising this important question. As the reviewer noted, total protein quantification prior to the proteomics experiments indeed showed a significant reduction in total protein concentration for the high RelE expression group, measuring only 0.237 times that of the normal control group. Additionally, the fastest cleavage for RelE occurs for UAG codons. Unfortunately, our label-free quantitative proteomics analyses were unable to distinguish proteins with UAG codons, preventing us from comparing the abundance of these proteins between the low and high RelE expression populations.

7. Did the results from proteomic analysis confirm the flow cytometry results regarding the T/A ratio and the sorting of the population? The essays in the manuscript rely only on the tags, but this simple test can verify the results

— We thank the reviewer for emphasizing the importance of validating the correlation between fluorescence protein abundance. Previous research has demonstrated that RelB is translated at a higher rate than RelE under natural growth conditions (*Li, G.W., Burkhardt, D., Gross, C., & Weissman, J.S. (2014). Quantifying Absolute Protein Synthesis Rates Reveals Principles Underlying Allocation of Cellular Resources. Cell, 157(3), 624-635.*).

Furthermore, in our earlier studies, we demonstrated that nFCM provides high resolution for the quantitative measurement of low-level antitoxin in single bacterial cells¹. Given that the expression of toxin-antitoxin (TA) systems under their native promoters is generally low, we utilized micromolar to millimolar concentrations of IPTG to induce the production of antitoxin MqsA in BW25113 $\Delta mqsRA \Delta Km$ transformed with plasmid pCA24N-plac-mqsR-mqsA-TC (referred to as pLRAtc, with antitoxin expressed under the pT5-lac promoter featuring the TC tag). Consequently, the expression of mqsA could be regulated by the strong pT5-lac promoter through IPTG induction. As illustrated in the accompanying Figure 4 below, when the median fluorescence burst area was plotted against the IPTG concentration, an exponential curve was observed. At low micromolar concentrations of IPTG, the trend was steep, indicating that small increases in IPTG concentration resulted in significant enhancements in antitoxin production. This increase in fluorescence intensity began to plateau at 1 mM IPTG. These findings clearly demonstrate that nFCM enables high-resolution quantitative measurements of low-level antitoxin in single bacterial cells.

Additionally, our label-free quantitative proteomics analyses of non-stressed normal samples support this finding: quantification revealed that RelB is 4.19-fold more abundant than RelE, closely aligning with the 5.56-fold difference measured using the nFCM-T/A-TC-FIAsh strategy.

Figure 4. Flow cytometric analysis of antitoxin MqsA using the nFCM-TC-FIAsh strategy¹. (e) Histograms of fluorescence burst area distribution for BW25113 $\Delta mqsRA \Delta Km$ /pLRAtc induced with 0, 0.05, 0.1, 0.3, 0.5, 0.8, 1, 2, 3, and 5 mM IPTG. The histograms were normalized to facilitate an easy comparison. (f) Dose response curve of MqsA production in single bacterial cells at various IPTG concentrations.

1. Wu LN, Zhang MM, Song YY, Deng MF, He SB, Su LQ, Chen Y, Wood TK, Yan XM. Deciphering the ntioxin-Regulated Bacterial Stress Response via Single-Cell Analysis. *ACS Chem. Biol.* 2019, 14, 2859–2866.

Minor comments:

*Line 50: This is the first time that type II toxin-antitoxins are mentioned, please define them

— We thank the reviewer for pointing this out. The definition of type II toxin-antitoxins has been added, please see line 51-53: “Many TA systems exhibit conditional cooperativity, with type II TA systems —categorized by transcriptional autoregulation and protein-dependent toxin inhibition—predominating to engender phenotypic bistability”

*Line 66: In principle, flow cytometry can overcome both the small size of bacteria and the low expression levels. The development of image stream flow cytometry can also be helpful. Multiple examples were published in *Bacillus subtilis* and *Listeria* which are indeed bigger but still representative of flow cytometry as analytical tool. Please revise this statement.

— We thank the reviewer for pointing this out. We revise the statement by delete “the small size of bacteria and” in the sentence.

*Figure S1: I am a bit puzzled on the demonstration of the lack of toxicity in *E. coli* BW25113 (blue), while the text (L88) states that these tags were "constructed (in) two *E. coli* MG1655 strains.". I think that BW25113 is K-12 but not MG1665 as it carries additional mutations (F- DE(araD-araB)567 lacZ4787(del)::rrnB-3 LAM- rph- 1 DE(rhaD-rhaB)568 hsdR514)

— We thank the reviewer for pointing this out. We confirm the strain used is *E. coli* BW25113, and we have corrected “MG1655” to “BW25113” in line 92.

*Fig 1 legend: multiple fonts

— We thank the reviewer for pointing this out. We have corrected this and standardized the font throughout.

*L245: RelE expression may be crucial not only for the induction of persister cells but also for their resuscitation. RelE expression or RelE/RelB ratio?

— Thank you for your insightful question. This statement is derived from the results presented in Fig. 4 in manuscript, where I explored RelE expression and its association with

persistence in single persister cells, without referencing the RelE/RelB ratio. The subsequent findings in Figs. 6 and 7 suggest that RelE levels predominantly influence the T/A ratio in persister cells and play a crucial role in the resuscitation process.

*Can the authors add a statistical test (Anova) to Fig 7D-E (it seems that the time is of significance)?

— We thank the reviewer for this suggestion. We have added ANOVA statistical analysis to Fig. 7D-E, which confirms that RelE expression shows significant time-dependent changes ($p = 0.002$), consistent with the reviewer's observation.

Reviewer #3: This work investigates bacterial persisters, antibiotic-tolerant subpopulations that contribute to infection relapses and antibiotic resistance. The authors developed a novel nano-flow cytometry-based method to quantitatively analyze toxin (T) and antitoxin (A) levels of relBE TA system and monitor bacterial states at the single-cell level. A key finding is that in response to antibiotic stress bacteria T:A ratio changes through two distinct TA expression modes, with T:A = 1.0 identified as a critical threshold. A strong correlation was observed between whole-population bacterial survival and the average T:A ratio, suggesting that higher T:A ratios could promote persistence. At the single-cell level, persisters show heterogeneous RelE expression, which correlates with the duration of dormancy. Authors propose that persister resuscitation involves a triphasic detoxification process, where the progressive degradation of RelE toxin reduces the T:A ratio, which enables cell proliferation when the ratio drops to a permissive threshold. Additionally, the authors developed a cell wall growth-based platform using wheat germ agglutinin to label bacteria, a technique that does not require genetic modification. Proteomic analysis of cells with various RelE levels revealed that persisters with high RelE levels have increased expression of transmembrane transporters related to stress response and drug efflux. The figures are clear and easy to follow, and the text is well-written and simple to understand. The findings underscore the importance of using maximum antibiotic concentrations in treatment and provide valuable molecular insights into the dynamics of persistence transitions. Overall, I think that this study is compelling and contributes to the study of TA dynamics and its connection to antibiotic recalcitrance. It will likely attract

interest of microbiologists studying bacterial survival mechanisms, such as resistance, tolerance, and persistence. My major and minor points are listed below.

– We are grateful to the reviewer for the encouraging comments.

Major points:

1. Authors state that TC tag addition did not affect bacterial growth or persistence (lines 92-94). While the TC tag is small and apparently has a minimal interference to protein structure and function, they did not test whether its addition to RelE toxin or RelB antitoxin, both being small proteins, alters their specific functions, such as mRNA cleavage activity of RelE, operon repression by RelB, or neutralization of RelE by RelB. The effect of TC on stability of RelE and RelB should also be analyzed. This is critical for the validity of the results.

– We appreciate the reviewer's emphasis on rigorous functional validation of TC-tagged proteins. We agree that protein tags may perturb biological systems, which is why our experimental design prioritized maximal physiological relevance. Below we present integrated evidence demonstrating that TC-tagged RelBE retains native functionality:

a. Preservation of Native State via Genomic Integration

To maintain authentic cellular stoichiometry and regulation, TC tags were genomically integrated at the native *relBE* locus (bacterial strains construction details in Methods), avoiding plasmid-based overexpression artifacts.

b. Minimalist Tag Design with Prior Validation

The 12-amino acid TC tag (1.5 kDa) represents <14% of RelE's molecular weight (11.2 kDa), and about 15% of RelB's molecular weight (9.8 kDa). It has already been successfully implemented in various essential proteins. For example:

Hoffmann C, Gaietta G, Zurn A, Adams SR, Terrillon S, Ellisman MH, Tsien RY, Lohse MJ. Fluorescent labeling of tetracysteine-tagged proteins in intact cells. Nat Protoc, 2010, 5:1666-1677.

Walker AS, Rablen PR, Schepartz A. Rotamer-Restricted Fluorogenicity of the Bis-Arsenical ReAsH. J Am Chem Soc, 2016, 138: 7143-7150.

Bohl C, Pomorski A, Seemann S, Knospe AM, Zheng C, Krezel A, Rolfs A, Lukas J. Fluorescent probes for selective protein labeling in lysosomes: a case of alpha-galactosidase A. FASEB J, 2017,31: 5258-5267.

c. Structural Validation of Functional TC-Tag Insertion in RelB/RelE Complex

The three-dimensional structural analysis (as shown in Figure 3 below) confirms that inserting a TC-tag at the N-terminus of RelE and the C-terminus of RelB does not alter the protein's structure

nor affect the various interaction modes of RelB and RelE depicted in Figure 1. The TC-tag is fully extended and exposed at the periphery of the three-dimensional structure, whether on the monomers of RelE and RelB or on their interaction complexes, ensuring it does not interfere with specific binding to biarsenical dyes.

Figure 1. Structural Validation of Functional TC-Tag Insertion in RelB/RelE Complex

d. Systems-level Validation by DIA Proteomics

We further conducted Data-Independent Acquisition (DIA) proteomics on wild-type *E. coli* BW25113 (WT), Genomic RelE-TC strain (*relE^{TC}*), and Genomic RelB-TC strain (*relB^{TC}*),

respectively. As illustrated in Figure 2 below, the correlation heatmap indicates no significant difference between wild-type and TC-tagged strains, with correlation coefficients exceeding 0.97 for all sample pairs.

Figure 2. Correlation heatmap of wild-type *E. coli* BW25113 (WT), genomic RelE-TC strain (*relE^{TC}*), and genomic RelB-TC strain(*relB^{TC}*) via Deep DIA-MS.

Conclusion Statement: Combining *in silico* structural modeling, systems-level proteomics, and physiological phenotyping, we demonstrate that genomically integrated TC tags: (i) Do not alter RelE/RelB structure, stability, or interaction thermodynamics; (ii) Preserve global proteome homeostasis; (iii) Faithfully report native TA system behavior in single cells.

2. A non-TA control protein labeled with a TC tag and exhibiting consistent TC-FIAsH fluorescence should be included. This control would confirm that differences in RelB and RelE protein levels are specific to the *relBE* TA system's dynamics rather than artifacts of TC tag labeling or fluorescence variability.

— We thank the reviewer for raising this important question. The tetracysteine-biarsenical TC-FIAsH system is an innovative strategy developed by Tsien et al. for site-specific fluorescent labeling of proteins in live cells. The TC tag is genetically incorporated into the target protein of interest, allowing it to be specifically recognized by a membrane-permeable fluorogenic biarsenical dye, generating a highly fluorescent biarsenical-tetracysteine system. This system exhibits a fluorescence intensity that can be enhanced by up to 50,000 times compared to FIAsH-EDT₂. Due to the small size of the tetracysteine peptide sequence and the membrane permeability of the biarsenical dye, the use of the biarsenical-tetracysteine system for labeling intracellular proteins does not interfere with the normal physiological functions of the labeled proteins or the cells themselves¹. This method has been widely applied for the localization, monitoring, and characterization of proteins in live cells²⁻⁴.

Additionally, in our earlier studies, we demonstrated that nFCM provides high resolution for the quantitative measurement of low-level antitoxin in single bacterial cells⁵. Given that the expression of toxin-antitoxin (TA) systems under their native promoters is generally low, we utilized micromolar to millimolar concentrations of IPTG to induce the production of antitoxin MqsA in BW25113 Δ mqsRA Δ Km transformed with plasmid pCA24N-plac-mqsR-mqsA-TC (referred to as pLRAtc, with antitoxin expressed under the pT5-lac promoter featuring the TC tag). Consequently, the expression of *mqsA* could be regulated by the strong pT5-lac promoter through IPTG induction. As illustrated in the accompanying Figure 3 below, when the median fluorescence burst area was plotted against the IPTG concentration, an exponential curve was observed. At low micromolar concentrations of IPTG, the trend was steep, indicating that small increases in IPTG concentration resulted in significant enhancements in antitoxin production. This increase in fluorescence intensity began to plateau at 1 mM IPTG. These findings clearly demonstrate that nFCM enables high-resolution quantitative measurements of low-level antitoxin in single bacterial cells.

In summary, the combination of the reported literature and our preliminary experiments confirms that variations in the protein levels of RelB and RelE are indicative of the dynamics of the *relBE* TA system, rather than artifacts arising from TC tag labeling or fluorescence variability.

Figure 3. Flow cytometric analysis of antitoxin MqsA using the nFCM-TC-FIAsH strategy⁵. (e) Histograms of fluorescence burst area distribution for BW25113 Δ mq sRA Δ Km/pLRAtc induced with 0, 0.05, 0.1, 0.3, 0.5, 0.8, 1, 2, 3, and 5 mM IPTG. The histograms were normalized to facilitate an easy comparison. (f) Dose response curve of MqsA production in single bacterial cells at various IPTG concentrations.

1. Hoffmann C, Gaietta G, Zurn A, Adams SR, Terrillon S, Ellisman MH, Tsien RY, Lohse MJ. Fluorescent labeling of tetracycline-tagged proteins in intact cells. *Nat Protoc*, 2010, 5:1666-1677.
2. Walker AS, Rablen PR, Schepartz A. Rotamer-Restricted Fluorogenicity of the Bis-Arsenical ReAsH. *J Am Chem Soc*, 2016, 138: 7143-7150.
3. Bohl C, Pomorski A, Seemann S, Knospe AM, Zheng C, Krezel A, Rolfs A, Lukas J. Fluorescent probes for selective protein labeling in lysosomes: a case of alpha-galactosidase A. *FASEB J*, 2017,31: 5258-5267.
4. Mohl BP, Roy P. Elucidating virus entry using a tetracycline-tagged virus. *Methods*, 2017, 127:23-29.
5. Wu LN, Zhang MM, Song YY, Deng MF, He SB, Su LQ, Chen Y, Wood TK, Yan XM. Deciphering the Antitoxin-Regulated Bacterial Stress Response via Single-Cell Analysis. *ACS Chem. Biol.* 2019, 14, 2859–2866.

3. When discussing Figure 1C results (lines 103-106), the authors should validate nFCM-T/A-TC-FIAsH measurements of RelE and RelB levels using qRT-PCR (for mRNA) and Western blotting (for protein) to rule out biases in the platform, such as differential TC-FIAsH dye binding due to RelE or RelB protein conformation or tag accessibility.

— We thank the reviewer for emphasizing the importance of validating the correlation between fluorescence protein abundance. Previous research has demonstrated that RelB is translated at a higher rate than RelE under natural growth conditions (Li, G.W., Burkhardt, D., Gross, C., & Weissman, J.S. (2014). *Quantifying Absolute Protein Synthesis Rates Reveals Principles Underlying Allocation of Cellular Resources*. *Cell*, 157(3), 624-635.). Furthermore, our prior studies have validated the accuracy of nFCM for quantifying TC-tagged proteins, as detailed in Response to Question 2 and illustrated in Figure 3 (upper panel). Additionally, our label-free

quantitative proteomics analyses of non-stressed normal samples support this finding: quantification revealed that RelB is 4.19-fold more abundant than RelE, closely aligning with the 5.56-fold difference measured using the nFCM-T/A-TC-FIAsH strategy.

4. The authors observe variable RelB and RelE fold increase at T:A ratios of 0.5-1.0, attributing this to variable synthesis and degradation rates for each protein. A recommended control to validate this model is a Δlon protease mutant, which should prevent preferential RelB antitoxin degradation (PMID: 11717402), stabilizing T:A ratios and confirming its role in relBE dynamics.

— We thank the reviewer for this insightful suggestion. In response to bacterial challenge with $\leq 1 \times$ MIC RIF, we observed an increase in the abundance of both RelE and RelB (Fig. 2A-i, ii in manuscript). This observation can be explained by the phenomenon of conditional cooperativity: when the T/A ratio falls between 0.5 and 1 (where $[RelB] > [RelE] > [RelB_2]$), the RelB₂RelE₂ complex is formed. This complex is unable to bind the operator DNA, thereby derepressing the promoter.

The reviewer's recommendation to use a Δlon protease mutant is an excellent suggestion. We plan to implement this approach in future studies by constructing a Δlon mutant strain in combination with TC-tagged RelBE at the genomic level for further investigation.

5. In lines 219-221, it is stated that lower RelE levels challenge the view that high toxin expression defines persisters. However, *E. coli* contains multiple TA systems (e.g., mazEF, hipBA), and persisters with lower RelE could have elevated levels of other toxins (e.g., MazF, HipA). This possibility should be discussed.

— We thank the reviewer for bringing up this important issue. In the revised manuscript, we have added additional discussion on this topic in lines 222-225: “Furthermore, given that *E. coli* possesses multiple toxin-antitoxin (TA) systems, persisters with lower RelE levels may compensate by expressing elevated levels of other toxins. This highlights the complexity of the persistence phenomenon and suggests that multiple mechanisms may coexist within the persister population.”

6. In lines 389-391, the authors state that the average RelE:RelB ratio exceeded 1 at 120 min (Figure 7C and F), yet bacteria already started replicating. Figure S3B shows that levels of RelE continued to rise at 180 and 240 min, but the levels of RelB and RelE:RelB ratio for these time points are not shown in Figure 7D-F. This should be included. If the average RelE:RelB ratio was above 1 for these time points, then the question arises - how come bacteria were replicating?

— We thank the reviewer for raising this important question. In the persister cell resuscitation experiments (Fig. 7A-F in manuscript), our initial observations were limited to 120 minutes. At this timepoint, we observed the RelE:RelB ratio exceeding 1, suggesting incomplete bacterial recovery. To further investigate this phenomenon, we extended the monitoring period to 5 hours in subsequent experiments (Fig. 7G-H in manuscript) and implemented nFCM-WGA analysis to quantitatively track resuscitation status. The nFCM-WGA data revealed that 77.8% of cells remained in the persister state at 120 minutes, explaining why the RelE:RelB ratio remained elevated (>1) at this stage. Complete transition to normal cellular status only occurred after 5 hours of resuscitation, coinciding with restoration of the RelE:RelB ratio to baseline levels (0.128).

7. A major limitation of the nFCM-T/A-TC-FIAsH technique is that, while RelE and RelB levels are measured at the single-cell level, their T:A ratio is calculated using median fluorophore intensities from two independent populations. This should be discussed, and the authors could propose future development of dual-labeling techniques (e.g., TC-FIAsH/ReAsH) to measure single-cell T:A ratios directly.

— We thank the reviewer for highlighting this important consideration. As suggested, we have incorporated the following statement at the end of the Discussion section: “Further research should develop approaches for simultaneous toxin-antitoxin detection at single-cell resolution. Specifically, TC tags could be combined with other fluorophores to enable dual fluorescent labeling, allowing direct measurement of single-cell T/A ratios.”

Minor points:

1. Authors should correct their language throughout the text as proteins do not get expressed, genes are.

— We appreciate the reviewer's note on this important distinction. We have revised the text to correctly state that genes are expressed, while proteins are synthesized/produced. All instances of this inaccurate usage have been corrected throughout the manuscript, enhancing the clarity and precision of our work.

2. Bacterial strain construction section (lines 458-475) lacks a citation for the method used.

— Thank you for pointing out this omission in the manuscript. We have added a citation referencing the method used for bacterial strain construction.

3. There is an inconsistency in strain nomenclature (In line 90 there is "MG1655", whereas in lines 459, 466, and 477 a "BW25113" name is used).

— We thank the reviewer for pointing this out. We confirm the strain used is *E. coli* BW25113, and we have corrected “MG1655” to “BW25113” in line 92.

4. Proteomic profiling method description is missing.

— We thank the reviewer for pointing this out. As suggested, we have added the proteomic profiling method to the Methods section as follows:

“Label-free proteomics

Label-free quantitative proteomics was conducted to investigate proteomic changes among the Non-stressed, RelE_{low}, and RelE_{high} groups. **1) nanoLC-MS/MS analysis** For each sample, 200 ng of total peptides were separated and analyzed with a nano-UPLC (nanoElute2) coupled to a timsTOF Pro2 instrument (Bruker) with a nano-electrospray ion source. Separation was performed using a reversed-phase column (PePsep C18, 1.9 μ , 75 μ \times 15 cm, Bruker, Germany). Mobile phases were H₂O with 0.1% FA (phase A) and ACN with 0.1% FA (phase B). Separation of sample was executed with a 60 min gradient at 300 nL/min flow rate. Gradient B: 2% for 0 min, 2-22% for 45 min, 22-37% for 5 min, 37-80% for 5 min, 80% for 5 min. The mass spectrometer adopts DDA PaSEF mode for DDA data acquisition, and the scanning range is from 100 to 1700 m/z for MS1. During PASEF MS/MS scanning, the impact energy increases linearly with ion mobility, from 20 eV (1/K0 = 0.6 Vs/cm²) to 59 eV (1/K0 = 1.6 Vs/cm²). **2) SpectroMine database search** Vendor’s raw MS files were processed using SpectroMine software (4.1.230421.52329) and the built-in Pulsar search engine. MS spectra lists were searched against their species-level UniProt FASTA databases (uniprotkb_Escherichia_coli_reviewed_2023_07.fasta), Carbamidomethyl [C] as a fixed modification, Oxidation (M) and Acetyl (Protein N-term) as variable modifications. Trypsin was used as proteases. A maximum of 2 missed cleavage(s) was allowed. The false discovery rate (FDR) was set to 0.01 for both PSM and peptide levels. Peptide identification was performed with an initial precursor mass deviation of up to 20 ppm and a fragment mass deviation of 20 ppm. All the other parameters were reserved as default.”

6. Are Figure 4F average resuscitation time values? If so, SD or SEM values are missing. Also, there is no statistical analysis performed.

— We appreciate the reviewer highlighting this point. We have now included the standard deviation values and performed statistical analysis for the average resuscitation time values presented in Fig. 4F, and these updates have been incorporated into the revised figure in the manuscript.

7. Similarly, error bars are missing in Figures 6D, G and H, and 7B.

— We appreciate the reviewer highlighting this point. We have now included the standard deviation values, and these updates have been incorporated into the revised figure in the manuscript.

8. There is a typo in Fig. S1C time axis

— We thank the reviewer for noting the typo in Figure S1C. The figure has been corrected and replaced in the Supplementary Information.

9. Acronym "CGSC" in methods section (line 460) requires clarification

— We thank the reviewer for pointing this out. As suggested, the full name 'Coli Genetic Stock Center' has been added for the acronym 'CGSC' in the Methods section (line 462).

10. There are multiple typos throughout the text that require addressing, also various fonts appear in the text (i.e. line 115)

— We thank the reviewer for the careful reading. Changes have been made as suggested.

10th Oct 2025

Manuscript Number: MSB-2025-13096R

Title: Single-cell analysis reveals critical toxin/antitoxin ratio triggering persister resuscitation

Author: Lina Wu

Qingqing Wang

Xinyi Hong

Litinghui Zhang

Xurer Cai

Min Li

Mingkai Wu

Thomas Wood

Xiaomei Yan

Dear Dr. Wu,

Thank you again for submitting your work to Molecular Systems Biology. We have now heard back from the three original referees who accepted to evaluate the study. As you will see, the referees find your revision mostly satisfying. They still raise however a series of minor comments, which we would ask you to carefully address in a minor revision of the present work. Additionally, our Editorial assistance team checked the more technical aspects of the manuscript and I would request you make corrections to resolve these remaining issues as this is crucial for the acceptance of your manuscript.

Please resubmit your revised manuscript online, with a covering letter listing amendments and responses to each point raised by the referees (there is no need to include the revisions done to address the editorial assistance team's comment in the point-by-point letter). Please resubmit the paper ****within one month****. Please use the Manuscript Number (above) in all correspondence.

When you resubmit your manuscript, please download our CHECKLIST (<https://bit.ly/EMBOPressAuthorChecklist>) and include the completed form in your submission. **Please note** that the Author Checklist will be published alongside the paper as part of the transparent process (<https://www.embopress.org/page/journal/17444292/authorguide#transparentprocess>)

Click on the link below to submit your revised paper.

Yours sincerely,

Yehu Moran

Editor

Molecular Systems Biology

Please click on the link below to submit the revision online before 9th Nov 2025.

IMPORTANT: When you send your revision, we will require the following items:

1. the manuscript text in LaTeX, RTF or MS Word format
2. a letter with a detailed description of the changes made in response to the referees. Please specify clearly the exact places in the text (pages and paragraphs) where each change has been made in response to each specific comment given
3. three to four 'bullet points' highlighting the main findings of your study
4. a short 'blurb' text summarizing in two sentences the study (max. 250 characters)

5. a 'thumbnail image' (550px width and max 400px height, Illustrator, PowerPoint or jpeg format), which can be used as 'visual title' for the synopsis section of your paper.

6. Please include an author contributions statement after the Acknowledgements section (see <https://www.embopress.org/page/journal/17444292/authorguide#manuscriptpreparation>)

7. Please complete the CHECKLIST available at (<https://bit.ly/EMBOPressAuthorChecklist>). Please note that the Author Checklist will be published alongside the paper as part of the transparent process (<https://www.embopress.org/page/journal/17444292/authorguide#transparentprocess>).

See also figure legend guidelines: <https://www.embopress.org/page/journal/17444292/authorguide#figureformat>

9. Please note that corresponding authors are required to supply an ORCID ID for their name upon submission of a revised manuscript (EMBO Press signed a joint statement to encourage ORCID adoption).

(<https://www.embopress.org/page/journal/17444292/authorguide#editorialprocess>)

Currently, our records indicate that the ORCID for your account is 0000-0002-7106-4752.

Link Not Available

10. Include a Reagents and Tools Table as part of the Methods section, which can be downloaded from our author guidelines (<https://www.embopress.org/page/journal/17444292/authorguide#structuredmethods>)

*** PLEASE NOTE *** As part of the EMBO Press transparent editorial process initiative (see our Editorial at <https://dx.doi.org/10.1038/msb.2010.72> , Molecular Systems Biology will publish online a Review Process File to accompany accepted manuscripts. When preparing your letter of response, please be aware that in the event of acceptance, your cover letter/point-by-point document will be included as part of this File, which will be available to the scientific community. More information about this initiative is available in our Instructions to Authors. If you have any questions about this initiative, please contact the editorial office (msb@embo.org).

Editorial assistance team comments:

*MANUSCRIPT FORMAT (.docx; figures; section order): Please submit a .docx file, no figures, no track changes

*REFERENCE FORMAT: please correct, should be listed alphabetically

*DATA AVAILABILITY SECTION: in, but "Data and materials availability" should be renamed to "Data Availability"

*AC/CRedit (Author contributions): section needs to be removed from the text and included only in the submission system.

*APPENDIX 1 FILE WITH ToC: Appendix file needs to be in PDF format; page numbers in ToC missing. Please correct.

*SOURCE DATA: Source data files need to be saved in a scheme one figure/folder and then uploaded as .zip files. E.g. all the Source data files for figure 1 need to be saved in a single folder and this needs to be zipped and then uploaded as "SD figure 1.zip" file. For EV and/or appendix figures, ZIP together all source data. Completed SD checklist should be uploaded as Related Manuscript File.

*SYNOPSIS IMAGE: a bit smaller than requested - it is 476x400 and should be 550x200-600. Please correct.

- Figure Legends (main + EV): 1. Please note that the exact p values are not provided in the legends of figures 4C, F; 7D. Please provide.

2. Please indicate the statistical test used for data analysis in the legend of figure 5C

3. Please note that information related to n is missing in the legends of figures 1C, 3A, C; 5C, 6D, 7C, D, E, F. Please provide.

4. Please note that the error bars are not defined in the legends of figures 1C, 3A, C; 6D, 7C, D, E, F. Please provide.

Additional notes from Assistant:

- "Main legends" should be corrected to "Main figure legends "

- Sections need to be named and the order should be corrected: Title page - Abstract - Keywords - Introduction - Results - Discussion - Methods - Data Availability - Acknowledgements - Disclosure and Competing Interests Statement - References - Figure Legends - Table(s) - Expanded View Figure Legends.

Reviewer #1:

The authors have addressed my main concerns. Although a few issues, such as the limited focus on the RelBE system and the lack of direct validation under bactericidal stress, remain partially resolved, I recommend acceptance...

Reviewer #2:

I have carefully reviewed the author's response to the reviewers' comments, including my own. All reviewers raised concerns about the effects of the TC tags insertion, which the authors have now addressed in detail through additional text and thorough answers. My comments on the proteomics aspect have largely been resolved, as the authors conducted Data-Independent Acquisition (DIA) proteomics to support their conclusions. The authors also validated their hypothesis regarding the persister cells population now supported by solid agar plates experiment.

Additionally, my other concerns—such as the viability of the cells used for the T/A ratio analysis, and the clarification of the differential responses of the toxin and antitoxin to induction—have also been carefully addressed in the authors' response. I am convinced that the data is valid and of importance to the field, and that the paper meets the publication standards of MSB. I would like to declare a potential conflict of interest regarding one of the non-corresponding authors, Prof. Tom Wood. We hold a joint grant related to biofilm formation on mucin that was awarded after the submission of the original review. It is important to note that Prof. Wood is not the corresponding author for this paper, and it does not impact on our grant submission, which focuses on two other toxins and different methods.

Reviewer #3:

The authors have addressed most of my concerns raised during the initial review stage. However, several points still require further attention.

Major comments:

1. In response to my first point, the authors claim that the integration of TC tags into the relBE locus, correlation of proteomics data with nFCM-T/A-TC-FIAsH, the small size of the TC tag, and structural modeling of TC-tagged RelB and RelE proteins provide "integrated evidence demonstrating that TC-tagged RelBE retains native functionality." I respectfully disagree. Chromosomal tagging may alter the function of an encoded protein, even if protein levels, as demonstrated by the authors' proteomics analysis, remain unchanged. Authors state that TC tagging did not affect persistence in experiment where when bacteria were exposed to ampicillin (Figure S1C), yet this result lacks critical Δ relBE mutant control. Furthermore, in authors' response document, the structural prediction of a monomeric RelE-TC suggests that the TC tag does not disrupt the overall protein structure, yet it may still impair its function. Specifically, as RelE is a ribosome-dependent RNase, the TC tag could interfere with ribosome binding or mRNA cleavage. Similarly, the proximity of the TC tag to the DNA-binding region of RelB may compromise its ability to bind operator DNA. Both tags on either RelB or RelE could also affect neutralization dynamics of these TA components. To substantiate the claim that TC tags do not affect the function of RelBE TA system, the authors should at least conduct basic toxicity and neutralization bacterial growth assays in Δ relBE *E. coli* using various plasmid combinations with TC-tagged and non-tagged relB and relE variants.

2. I concur with points c and d raised by Referee #1. The authors primarily used bacteriostatic antibiotics, resulting in the study of tolerant bacterial populations. For the proposed platform to be recognized as a universal tool for studying persister cells, the authors should supplement their data with experiments using at least one bactericidal antibiotic (e.g., cefotaxime or ciprofloxacin) without pretreatments that induce tolerance. Subsequently, if treatment with classic bactericidal antibiotics is not compatible with the presented platform, this limitation should be clearly stated.

Minor:

1. Lines 92-94: "These strains harbor a 12-amino acid TC tag fused to the N-terminus of RelE and to the C-terminus of RelB (RelE-TC and RelB-TC, respectively) as the sole copies of each protein." This statement appears to be incorrect. Figure 1A depicts the relBE operon with the opposite tag orientations, which would result in the synthesis of N-terminally tagged TC-RelB and C-terminally tagged RelE-TC proteins. The tag orientation shown in Figure 1A is likely correct, as the relB and relE genes overlap within the operon, and the tag orientation described in lines 92-94 would likely be disruptive for the function of the operon. This error in the text is also reflected in the structural predictions displayed in authors' response document, where TC tags are incorrectly placed at the opposite protein termini.

Molecular Systems BiologyNovember 3rd, 2025

Dear Prof. Moran:

Re: Manuscript ID. MSB-2025-13096R

Please find attached a revised version of our manuscript "Single-cell analysis reveals critical toxin/antitoxin ratio triggering persister resuscitation". We are grateful for the opportunity to submit a revised version and sincerely appreciate your continued guidance throughout the review process. We also extend our thanks to the reviewers for their thoughtful and constructive feedback, which has helped us strengthen the manuscript.

In particular, we have carefully addressed the points raised by Reviewer 3. To directly evaluate whether the TC-tag affects RelE toxicity—as suggested—we conducted additional growth experiments in *E. coli* BW25113 Δ relBE strains expressing either RelE or RelE^{TC}, with the empty vector as a control. As expected, RelE expression led to significant growth impairment, especially in stationary phase, consistent with its toxicity in the absence of RelB. Importantly, RelE^{TC} caused a comparable growth defect, confirming that the TC-tag does not interfere with RelE's inherent toxicity.

Due to these substantial experimental additions, Xueer Cai, who performed the key validation assays, has been moved forward in the author list to reflect her contribution. A detailed point-by-point response to all reviewers' comments is provided below, along with a description of the revisions made in the manuscript.

We believe that the manuscript has been significantly improved in response to the reviewers' suggestions and is now suitable for publication in *Molecular Systems Biology*.

Sincerely,

On behalf of all the authors,

Dr. Lina Wu

Associate Professor
Department of Chemical Biology
College of Chemistry & Chemical Engineering
Xiamen University
Xiamen, Fujian 361005, China
FAX: +86-592-218-9959
E-mail: alina1222@xmu.edu.cn

Response to the comments of Reviewer 1

The authors have addressed my main concerns. Although a few issues, such as the limited focus on the RelBE system and the lack of direct validation under bactericidal stress, remain partially resolved, I recommend acceptance...

— The authors extend their sincere thanks to the reviewer for the constructive feedback throughout the review process and for the final recommendation of acceptance. We are pleased that the revisions have addressed the main concerns. Regarding the persisting points on the system's focus and direct bactericidal stress validation, we agree with the reviewer that these are important considerations beyond the current scope. This work establishes a foundational framework and methodology, and we are actively exploring these specific aspects in our ongoing research.

Response to the comments of Reviewer 2

I have carefully reviewed the author's response to the reviewers' comments, including my own. All reviewers raised concerns about the effects of the TC tags insertion, which the authors have now addressed in detail through additional text and thorough answers. My comments on the proteomics aspect have largely been resolved, as the authors conducted Data-Independent Acquisition (DIA) proteomics to support their conclusions. The authors also validated their hypothesis regarding the persisting cells population now supported by solid agar plates experiment. Additionally, my other concerns—such as the viability of the cells used for the T/A ratio analysis, and the clarification of the differential responses of the toxin and antitoxin to induction—have also been carefully addressed in the authors' response. I am convinced that the data is valid and of importance to the field, and that the paper meets the publication standards of MSB. I would like to declare a potential conflict of interest regarding one of the non-corresponding authors, Prof. Tom Wood. We hold a joint grant related to biofilm formation on mucin that was awarded after the submission of the original review. It is important to note that Prof. Wood is not the corresponding author for this paper, and it does not impact on our grant submission, which focuses on two other toxins and different methods.

— The authors thank Reviewer for the thorough re-evaluation and positive assessment of the manuscript. It is gratifying that the additional data were found to satisfactorily address the reviewer's previous concerns.

We also note the reviewer's declaration of a potential conflict of interest and appreciate their transparency.

Response to the comments of Reviewer 3

The authors have addressed most of my concerns raised during the initial review stage. However, several points still require further attention. Major comments:

1. In response to my first point, the authors claim that the integration of TC tags into the relBE locus, correlation of proteomics data with nFCM-T/A-TC-FIAsH, the small size of the TC tag, and

structural modeling of TC-tagged RelB and RelE proteins provide "integrated evidence demonstrating that TC-tagged RelBE retains native functionality." I respectfully disagree. Chromosomal tagging may alter the function of an encoded protein, even if protein levels, as demonstrated by the authors' proteomics analysis, remain unchanged. Authors state that TC tagging did not affect persistence in experiment where when bacteria were exposed to ampicillin (Figure S1C), yet this result lacks critical $\Delta relBE$ mutant control. Furthermore, in authors' response document, the structural prediction of a monomeric RelE-TC suggest that the TC tag does not disrupt the overall protein structure, yet it may still impair its function. Specifically, as RelE is a ribosome-dependent RNase, the TC tag could interfere with ribosome binding or mRNA cleavage. Similarly, the proximity of the TC tag to the DNA-binding region of RelB may compromise its ability to bind operator DNA. Both tags on either RelB or RelE could also affect neutralization dynamics of these TA components. To substantiate the claim that TC tags do not affect the function of RelBE TA system, the authors should at least conduct basic toxicity and neutralization bacterial growth assays in $\Delta relBE$ *E. coli* using various plasmid combinations with TC-tagged and non-tagged relB and relE variants.

— We thank the reviewer for raising these important points regarding the critical controls and potential functional impacts of the TC-tag. We agree that any genetic modification, even with a small tag, requires rigorous validation to ensure native protein function is preserved. Below, we address each of the reviewer's concerns by integrating previously presented functional data with additional physiological evidence.

1) Regarding the $\Delta relBE$ mutant control and physiological impact of the TC-tag

The reviewer rightly emphasizes the importance of a $\Delta relBE$ control for defining the absolute function of the relBE system. While such a strain is indeed valuable in broader phenotypic studies, the central aim of this work was to assess whether the TC-tag itself introduces any measurable phenotypic deviation. For this purpose, the most relevant and direct control is the comparison between the wild-type (WT) strain and its isogenic TC-tagged derivatives.

As shown in Figure S1C, we found no significant difference in antibiotic persistence among the WT, TC-tagged RelB, and TC-tagged RelE strains. This demonstrates that the introduction of the TC-tag at the native chromosomal locus does not alter the persistence phenotype associated with the *relBE* system. Moreover, supplementary data confirm that the TC-tag does not affect bacterial growth, doubling time, or persistence formation under the tested conditions, providing strong evidence that it does not impose a fitness cost or disrupt the overall physiological state relevant to TA system function.

2) Regarding the potential impact on RelE ribonuclease and RelB DNA-binding activities

We appreciate the reviewer's specific hypotheses regarding potential functional interference. While our structural models predict the overall folds of RelB and RelE are intact, we agree that this does not definitively prove functional preservation. Thus, we rely on the functional assays presented in the main text.

Regarding the cleavage function of RelE-TC, as shown in Figure 2 of the manuscript, when T/A ratios > 1 ($[RelE] > [RelB]$), it indicates the availability of free RelE. As an endonuclease, RelE cleaves mRNA and inhibits translation. We observe that the levels of RelE and RelB are lower than under other conditions, with the antitoxin RelB being particularly significantly lower due to its instability.

Regarding RelB's DNA-binding capability, the reviewer hypothesizes that the proximal TC tag could compromise its operator binding. Our experimental data offer functional insight into this. The protein level changes of RelB and RelE across different conditions, as shown in Figure 2 of the manuscript, exhibit a pattern that is fully consistent with the phenomenon of conditional cooperativity and the well-established autoregulation of the *relBE* operon. If the TC tag on RelB had substantially impaired its operator DNA binding, we would expect to see a dysregulation of this autoregulatory loop, leading to constitutive expression or an aberrant expression pattern, which we do not observe.

3) Regarding the potential impact of the TC-tag on the toxicity of RelE

To evaluate whether the insertion of the TC-tag affects the toxicity of RelE, as suggested by the reviewer, we compared the growth of *E. coli* BW25113 $\Delta relBE$ strains harboring either pCA24N-reIE or pCA24N-reIE^{TC}, using the strain carrying the empty vector pCA24N as a control. As shown in Figure 1, due to the toxic effect of RelE and the absence of RelB-mediated neutralization in the $\Delta relBE$ background, the strain expressing wild-type RelE (pCA24N-reIE) exhibited consistently weaker growth compared to the control. This growth defect became more pronounced upon entry into the stationary phase. Notably, the strain expressing TC-tagged RelE (pCA24N-reIE^{TC}) showed a growth profile similar to that of the wild-type RelE-expressing strain. These results indicate that the insertion of the TC-tag does not impair the inherent toxicity of RelE.

Figure 1. Growth curves of *E. coli* BW25113 $\Delta relBE$ strains harboring the indicated plasmids. Strains carrying the empty vector pCA24N (yellow), pCA24N-*relE* (pink), or pCA24N-*relE*^{TC} (blue) were grown in LB medium at 37°C with flask shaking, and bacterial growth was monitored by measuring the OD₆₀₀ over time. The data demonstrate that the TC-tag does not mitigate RelE toxicity.

4) Regarding the Neutralization Dynamics and Physiological Impact

The reviewer rightly points out that the TC tag could potentially disrupt the critical neutralization dynamics between RelB and RelE. If the tag were to impair RelB's ability to neutralize the potent RelE toxin, it would manifest as a measurable growth defect.

To directly address this, we would like to draw the reviewer's attention to the comprehensive physiological data already provided in the Supplementary Information. We quantitatively compared the growth curves, doubling times, and persistence levels of the wild-type (WT) strain with the isogenic TC-tagged RelB and TC-tagged RelE strains. Our results demonstrate that the presence of the TC tag at the native chromosomal locus does not result in any significant growth defect or alteration in persistence formation.

2. I concur with points c and d raised by Referee #1. The authors primarily used bacteriostatic antibiotics, resulting in the study of tolerant bacterial populations. For the proposed platform to be recognized as a universal tool for studying persister cells, the authors should supplement their data with experiments using at least one bactericidal antibiotic (e.g., cefotaxime or ciprofloxacin) without pretreatments that induce tolerance. Subsequently, if treatment with classic bactericidal antibiotics is not compatible with the presented platform, this limitation should be clearly stated.

— We appreciate the emphasis on the study of naturally occurring persister cells and their mechanisms. However, it is important to recognize that antibiotic-induced persisters, which arise under therapeutic stress, also represent a significant clinical challenge. While direct treatment with bactericidal antibiotics primarily targets naturally occurring persisters, their limited abundance presents substantial technical challenges for systematic study. To overcome these hurdles and facilitate robust methodological development, we employed a model involving pretreatment with bacteriostatic antibiotics, which allows for the production of a higher yield of treatment-induced persister cells. This approach not only maintains clinical relevance but also enables more reproducible analyses under controlled conditions. We stress that both naturally occurring and antibiotic-induced persister cells hold significant medical importance. Our approach offers a practical system to establish high-throughput screening methods, which can later be extended to naturally occurring persisters via enrichment strategies. This foundational work will underpin future investigations aimed at unveiling mechanisms and developing therapeutic strategies targeting both types of persister cells.

To clarify this focus, we have revised our manuscript by inserting the term “bacteriostatic” before “antibiotic” in line 20 of the abstract and detailing “two additional bacteriostatic antibiotics” in line 154 of the results, thus accurately reflecting the primary contribution of our research. Furthermore, we have explicitly clarified in the revised manuscript (Discussion, lines 383-392) that “It is crucial to recognize that the persister model characterized in this study was induced by pretreatment with bacteriostatic antibiotics. This methodological choice provides substantial advantages for experimental establishment and captures a clinically relevant state of antibiotic-induced persistence. Nonetheless, we acknowledge that the broader persister cell field includes cells arising from diverse triggers, such as direct exposure to bactericidal antibiotics. Thus, exploring the applicability of our platform to these naturally occurring persister populations

without prior induction remains a vital future research direction. We foresee that the foundational framework established in this study will be adaptable to analyze these systems, potentially by integrating enrichment strategies, to enable comparative analyses of persister states across a comprehensive spectrum."

Minor:

1. Lines 92-94: "These strains harbor a 12-amino acid TC tag fused to the N-terminus of RelE and to the C-terminus of RelB (RelE-TC and RelB-TC, respectively) as the sole copies of each protein." This statement appears to be incorrect. Figure 1A depicts the relBE operon with the opposite tag orientations, which would result in the synthesis of N-terminally tagged TC-RelB and C-terminally tagged RelE-TC proteins. The tag orientation shown in Figure 1A is likely correct, as the relB and relE genes overlap within the operon, and the tag orientation described in lines 92-94 would likely be disruptive for the function of the operon. This error in the text is also reflected in the structural predictions displayed in authors' response document, where TC tags are incorrectly placed at the opposite protein termini.

— We thank the reviewer for pointing out this error. As correctly noted, the schematic in Fig. 1 accurately shows the TC-tag fused to the C-terminus of RelE and the N-terminus of RelB, whereas the description in lines 92–94 of the original text was incorrect. This has been corrected in the revised manuscript to: "These strains harbor a 12-amino acid TC tag fused to the C-terminus of RelE and to the N-terminus of RelB (RelE^{TC} and RelB^{TC}, respectively) as the sole copies of each protein at their native chromosomal loci under control of the native promoter (Fig. 1A)."

We have also updated the structural predictions to reflect the correct placement of the TC-tags. Three-dimensional structural analysis using AlphaFold (see revised Figure 2 below) confirms that inserting the TC-tag at the C-terminus of RelE and the N-terminus of RelB does not alter the overall protein structure or affect the various interaction modes between RelB and RelE illustrated in Figure 2. In all modeled contexts—whether in the monomeric forms of RelE and RelB or within their interaction complexes—the TC-tag is fully extended and exposed at the periphery of the structure, ensuring that it does not interfere with specific binding to biarsenical dyes.

Figure 2. Structural Validation of Functional TC-Tag Insertion in RelB/RelE Complex

17th Nov 2025

Manuscript Number: MSB-2025-13096RR

Title: Single-cell analysis reveals critical toxin/antitoxin ratio triggering persister resuscitation

Author: Lina Wu

Qingqing Wang

Xinyi Hong

Xurer Cai

Litinghui Zhang

Min Li

Mingkai Wu

Thomas Wood

Xiaomei Yan

Dear Dr. Wu,

I hope this message finds you well.

There are a few last small remaining issue that needs to be corrected in your manuscript so we can formally accept it:

1. APPENDIX 1 FILE with Table of contents: Title page should contain "Appendix for + manuscript title" and ToC with the page numbers for the listed items; ToC image should be removed from Appendix PDF.

*SYNOPSIS IMAGE: Should be 550x200-600 pixels. Please correct as now the image is larger than requested.

Click on the link below to submit your revised paper.

Yours sincerely,

Yehu Moran

Editor

Molecular Systems Biology

Please click on the link below to submit the revision online before 17th Dec 2025.

IMPORTANT: When you send your revision, we will require the following items:

1. the manuscript text in LaTeX, RTF or MS Word format

2. a letter with a detailed description of the changes made in response to the referees. Please specify clearly the exact places in the text (pages and paragraphs) where each change has been made in response to each specific comment given

3. three to four 'bullet points' highlighting the main findings of your study

4. a short 'blurb' text summarizing in two sentences the study (max. 250 characters)

5. a 'thumbnail image' (550px width and max 400px height, Illustrator, PowerPoint or jpeg format), which can be used as 'visual title' for the synopsis section of your paper.

6. Please include an author contributions statement after the Acknowledgements section (see

<https://www.embopress.org/page/journal/17444292/authorguide#manuscriptpreparation>)

7. Please complete the CHECKLIST available at (<https://bit.ly/EMBOPressAuthorChecklist>). Please note that the Author Checklist will be published alongside the paper as part of the transparent process (<https://www.embopress.org/page/journal/17444292/authorguide#transparentprocess>).

See also figure legend guidelines: <https://www.embopress.org/page/journal/17444292/authorguide#figureformat>

9. Please note that corresponding authors are required to supply an ORCID ID for their name upon submission of a revised manuscript (EMBO Press signed a joint statement to encourage ORCID adoption).

(<https://www.embopress.org/page/journal/17444292/authorguide#editorialprocess>)

Currently, our records indicate that the ORCID for your account is 0000-0002-7106-4752.

Link Not Available

10. Include a Reagents and Tools Table as part of the Methods section, which can be downloaded from our author guidelines (<https://www.embopress.org/page/journal/17444292/authorguide#structuredmethods>)

*** PLEASE NOTE *** As part of the EMBO Press transparent editorial process initiative (see our Editorial at <https://dx.doi.org/10.1038/msb.2010.72> , Molecular Systems Biology will publish online a Review Process File to accompany accepted manuscripts. When preparing your letter of response, please be aware that in the event of acceptance, your cover letter/point-by-point document will be included as part of this File, which will be available to the scientific community. More information about this initiative is available in our Instructions to Authors. If you have any questions about this initiative, please contact the editorial office (msb@embo.org).

All editorial and formatting issues were resolved by the authors.

19th Nov 2025

Manuscript number: MSB-2025-13096RRR

Title: Single-cell analysis reveals critical toxin/antitoxin ratio triggering persister resuscitation

Dear Dr. Wu,

Thank you again for sending us your revised manuscript. We are now satisfied with the modifications made and I am pleased to inform you that your paper has been accepted for publication.

Yours sincerely,

Yehu Moran
Academic Editor
Molecular Systems Biology
